# Dynamical drivers of Greenland blocking in climate models

Clio Michel*[1], Erica Madonna*[1], Clemens Spensberger[1], Camille Li[1], and Stephen Outten[2]

[1]Geophysical Institute, University of Bergen and Bjerknes Centre for Climate Research, Bergen, Norway
[2]Nansen Environmental and Remote Sensing Center and Bjerknes Center for Climate Research, Bergen, Norway
*The authors equally contributed to the study

**Correspondence:** Clio Michel (Clio.Michel@uib.no)

**Abstract.** Blocking over Greenland is known to lead to strong surface impacts, such as ice sheet melting, and a change in its future frequency can have important consequences. However, as previous studies demonstrated, climate models underestimate the blocking frequency for the historical period. Even though some improvements have recently been made, the reasons for the model biases are still unclear. This study investigates whether models with realistic Greenland blocking frequency in winter have a correct representation of its dynamical drivers, most importantly, cyclonic wave breaking (CWB). Because blocking is a rare event and its representation is model-dependent, we use a multi-model large ensemble. We focus on two models that show typical Greenland blocking features, namely a ridge over Greenland and an equatorward-shifted jet over the North Atlantic. ECHAM6.3-LR has the best representation of CWB of the models investigated, but only the second best representation of Greenland blocking frequency, which is underestimated by a factor of two. While MIROC5 has the most realistic Greenland blocking frequency, it also has the largest (negative) CWB frequency bias, suggesting that another mechanism leads to blocking in this model. Composites over Greenland blocking days show that the present and future experiments of each model are very similar to each other in both amplitude and pattern and that there is no significant change of Greenland blocking frequency in the future. However, these projected changes in blocking frequency are highly uncertain as long as the mechanisms leading to blocking formation and maintenance in models remain poorly understood.

## 1 Introduction

Blocking in the atmosphere is a persistent quasi-stationary anticyclonic anomaly that disrupts the westerly flow (Rex, 1950). Often occurring in mid-latitude regions such as Greenland and Europe/Scandinavia (Treidl et al., 1981; Davini et al., 2014), it has profound impacts on surface weather, leading to temperature extremes such as cold spells in winter (Trigo et al., 2004; Sillmann and Croci-Maspoli, 2009) and heat waves in summer (Pfahl and Wernli, 2012; Schaller et al., 2018). Blocking over Greenland is shown to last longer than blocking over other regions (e.g, Davini et al., 2012; Drouard et al., 2021). Moreover, Greenland blocking has been shown to cause melting events of the Greenland Ice Sheet (Fettweis et al., 2013; McLeod and Mote, 2015; Hermann et al., 2020; Hanna et al., 2021), by mainly reducing the cloud cover, increasing temperatures (Chen and Luo, 2017), and changing the surface energy budget (Ward et al., 2020, and references therein), which can impact global sea level rise (Van den Broeke et al., 2016). In addition to the local impact, Greenland blocking is also associated with temperature anomalies over the whole Northern Hemisphere (Chen and Luo, 2017). It is therefore crucial that models correctly represent

blocking in order to accurately simulate its impacts and potential future changes. Unfortunately, blocking frequency over the North Atlantic is still underestimated by the large majority of climate models despite some improvements in recent years (see Davini and D'Andrea, 2020, for a review). Moreover, as blocking events are also sporadic and exhibit a large natural variability (Woollings et al., 2018), some models will not simulate enough blocking events to robustly investigate the mechanisms leading to blocking, hence the need for very long simulations or many different realizations of the same experiment. In the present study, we make use of large ensembles of climate models with relatively coarse spatial resolutions that provide a broad sampling of the internal variability of the atmosphere to investigate the biases in Greenland blocking and the dynamical driving from Rossby wave breaking (RWB). Moreover, knowing these biases, we will look at how changes in RWB in a 2°C warmer world will shape future Greenland blocking frequency and pattern.

Climate models have steadily improved over the last decades but still struggle to correctly represent some important features of the atmospheric circulation such as the jet stream, the storm tracks, and the blocking over both the North Pacific and Atlantic. In the North Atlantic, the jet streams and storm tracks produced by the climate models from the various phases of the Coupled Model Intercomparison Project (CMIP) continue to be too zonal and placed too far south compared to reanalyses (Harvey et al., 2020). Most CMIP5 models do not reproduce the observed blocking frequency in the North Atlantic sector (Vial and Osborn, 2012; Masato et al., 2013; Anstey et al., 2013; Davini and D'Andrea, 2016) with up to a 30–50% underestimation of wintertime blocking frequencies (Woollings et al., 2018). Similar blocking biases are found in uncoupled climate models (e.g. Davini and D'Andrea, 2016). Many of the new generation models (CMIP6) show an improvement in reproducing blocking frequencies, but for some regions, such as the North Atlantic, most still have too little blocking (Davini and D'Andrea, 2020; Schiemann et al., 2020). Some studies have shown that statistically correcting the model's mean state improves the overall frequency of blocking and that a blocking detection method based on anomalies might be less sensitive to mean state biases compared to a method based on the meridional reversal of the geopotential gradient (Scaife et al., 2010; Vial and Osborn, 2012). However, Schiemann et al. (2020) showed that using an anomaly threshold does not necessarily remove the general blocking biases.

Many studies have documented model biases, but only some have tried to understand the physical drivers. For example, several studies have reported that biases in blocking are associated with biases in the mean flow (Scaife et al., 2010; Vial and Osborn, 2012) but have not further explored potential mechanisms linking the mean state to the occurrence of blocking. Other studies have reported a general decrease of North Atlantic blocking bias with increased model resolution (Anstey et al., 2013; Davini and D'Andrea, 2016; Davini et al., 2017; Schiemann et al., 2017; Davini and D'Andrea, 2020; Schiemann et al., 2020). With increased resolution comes a better representation of the orography, which in turn improves the mean state (e.g. the stationary wave patterns) and variability. Through this mechanism increased resolution can reduce blocking biases, but the benefits vary regionally (Berckmans et al., 2013; Davini et al., 2017). Similarly, Pithan et al. (2016) showed that a better parameterisation of orographic drag improved the blocking representation over the North Atlantic, but had the opposite effect over the North Pacific. Finally, Davini et al. (2017) found that realistic blocking frequencies may result from bias compensations: overly strong eddies at upper levels counterbalance the overly weak eddies at lower levels, with the higher-resolution models not necessarily better representing the eddies.

Eddy-mean flow interactions through the breaking of Rossby waves have been shown to be key for blocking onset and maintenance as they advect low-PV air towards the higher latitudes (Nakamura and Wallace, 1993; Pelly and Hoskins, 2003; Altenhoff et al., 2008; Tyrlis and Hoskins, 2008). The advection results in the formation of a ridge linked to an anticyclonic anomaly over Greenland. In addition, recent studies have shown that diabatic processes, such as the release of latent heat from rising air masses within extratropical cyclones, help amplify the ridge building and the formation and maintenance of blocking (Pfahl et al., 2015; Steinfeld and Pfahl, 2019). Cyclonic wave breaking on the poleward side of the North Atlantic jet (southwest of Greenland) precedes blocking over Greenland (Woollings et al., 2008; Michel and Rivière, 2011). During Greenland blocking, the North Atlantic jet is zonal and shifted southward compared to its climatological position (Woollings et al., 2008, 2010; Davini et al., 2014; Madonna et al., 2017). Kwon et al. (2018) analysed the daily jet variability in the Community Earth System Model version 1 Large Ensemble and documented an underestimation of Greenland blocking linked to the infrequent and non-persistent southward displacement of the North Atlantic eddy-driven jet. Similar results were found for other CMIP5 models. For example, Iqbal et al. (2018) showed that most models underestimate eddy-driven jet variability because of infrequent southward excursions of the North Atlantic jet.

While it is well documented that Greenland blocking frequency is underestimated in climate models, little is known about the representation of the key processes that lead to Greenland blocking. Therefore, the aim of the present study is to investigate the Greenland blocking frequency and pattern representation in climate models, as well as the dynamical processes leading to Greenland blocking, with a focus on RWB. We analyse five large ensembles (≥100 members) of atmosphere-only (AMIP) simulations, from the Half-a-degree Additional warming, Prognosis and Projected Impacts (HAPPI) project (Mitchell et al., 2017). With this large ensemble, we can assess the uncertainty in the representation of blocking due to internal atmospheric variability and the models themselves (structural uncertainty) and thus better evaluate the significance of biases. Finally, we look at how the frequency and dynamics of Greenland blocking may change in a $2°C$ warmer world relative to the pre-industrial period.

## 2 Data and Methods

### 2.1 The HAPPI large ensemble

The HAPPI international project provided a large multi-model ensemble with the aim to investigate the climate impacts in weak warming scenarios (Mitchell et al., 2017). In this study, we are interested in the present decade which covers 2006-2015 and a future decade in which the global annual mean temperature is $2°C$ warmer than the pre-industrial level ($\sim +1.2°C$ compared to the present decade). This multi-model ensemble comprises five atmospheric general circulation models (AGCM) with between 100 and 501 members for each period. For the present decade, the observed sea surface temperatures (SST) and sea ice were used. The greenhouse gases concentrations, aerosols, ozone, land use and land cover representative of 2006-2015 were held constant during the simulations. For the future decade, the CMIP5 ensemble mean SST and sea ice responses to global warming were added to the observed fields. More details about the simulations, models, and the forcings (atmospheric greenhouse gases, aerosols, ozone, and land) can be found in Mitchell et al. (2017) and Li et al. (2018). In the following, we

use the daily outputs of geopotential at 500 hPa, zonal and meridional wind at 850 and 250 hPa from the five models CAM4-2degree, CanAM4, ECHAM6.3-LR, MIROC5, and NorESM1-HAPPI. The horizontal and vertical resolutions as well as the number of members for each model can be found in the Appendix (Table S1). The shortcomings of using a relatively short simulation period (10 years) to investigate climate variability is compensated by the large number of members available, which allows for robust statistics. Moreover, Davini and D'Andrea (2016) did not find substantial differences in blocking statistics from climate models when using ten years compared to longer periods.

## 2.2 Reanalysis

We utilize the ERA-Interim reanalysis (Dee et al., 2011) from the European Centre for Medium-Range Weather Forecasts as a reference. Six-hourly data are averaged to produce daily means and interpolated horizontally to a $0.5° \times 0.5°$ grid. We consider the nine winter (December-January-February, DJF) seasons during the 2006-2015 decade (starting in December 2006 and ending in February 2015), which correspond to the decade used for the HAPPI simulations (see Sec. 2.1). We use the zonal and meridional components of the wind at lower (850 hPa) and upper (250 hPa) tropospheric levels, the geopotential height at 500 hPa, and the absolute vorticity at 250 hPa. In the remainder of the text, we use the time-mean 850 hPa zonal winds as a proxy for the eddy-driven jet. As the North Atlantic jet is predominantly eddy-driven (Eichelberger and Hartmann, 2007; Woollings et al., 2010; Li and Wettstein, 2012), we expect similar results if we use 250 hPa winds instead.

## 2.3 Blocking detection

Blocking refers to quasi-stationary and persistent weather systems that obstruct the westerly flow. There are many ways to identify blocking using anomalies or meridional gradients of various fields, such as 500-hPa geopotential, potential vorticity or temperature (e.g. Tibaldi and Molteni, 1990; Pelly and Hoskins, 2003; Scherrer et al., 2006; Davini et al., 2012; Masato et al., 2012; Dunn-Sigouin and Son, 2013), and each method has its own shortcomings (Tyrlis et al., 2020). In this study, we use reversals in the meridional gradient of the geopotential height at 500 hPa (Z500) to identify blocks (Tibaldi and Molteni, 1990). We follow the criteria of Scherrer et al. (2006) and detect blocking, lasting for at least 5 days, by looking for Z500 meridional gradient reversals in $30°$ latitudinal bands ($\pm 15°$) around every latitude between 35 and 75°N. Blocked grid points are identified from daily data for both ERA-Interim and the HAPPI simulations using the models' original grids and the whole decade. An interpolation to a common grid before identifying blocking does not lead to substantial changes in the results (see Fig. S1 for NorESM1-HAPPI). Winter-time blocking climatologies are obtained by averaging all blocked grid points over time (excluding January, February 2006 and December 2015 to only keep full DJF seasons) and are expressed as a percentage of the number of winter days ($90 \times 9 = 810$ days). Greenland blocking days are defined when at least 10% of the area within 65-25°W/60-75°N (black box in Fig. 1f) is blocked. Although 10% was subjectively chosen, it appears to capture relevant Greenland blocking days. The numbers of blocked days for each of the nine winters within the decade are averaged to give the mean DJF Greenland blocking frequency for each member. Composites of Greenland blocking are computed by averaging all days that exhibit blocking over Greenland.

## 2.4 Rossby wave breaking detection

Rossby wave breaking occurs when the waves elongate in a certain direction, break and dissipate. Anticyclonic wave breaking (AWB) occurs when the wave elongates along a northeast-southwest axis, typically on the equatorward flank of the jet, and acts to shift the eddy-driven jet poleward. Cyclonic wave breaking (CWB) occurs when the wave elongates along a northwest-southeast axis on the poleward flank of the jet to shift the eddy-driven jet equatorward. Here, we use the same detection algorithm as in Michel and Rivière (2011) based on the method of Rivière (2009) applied to the daily absolute vorticity fields interpolated on a regular $4.5° \times 4.5°$ spatial grid to capture the large-scale contour overturnings. The method identifies Rossby wave breaking via meridional reversals of absolute vorticity contours at 250 hPa. This method is known to provide similar statistics to those which use potential vorticity on different isentropic levels (Michel and Rivière, 2011; Barnes and Hartmann, 2012). This algorithm distinguishes between CWB and AWB by the direction of the contour reversal. Wave breaking frequencies are then derived by appropriately averaging over the binary mask fields. As CWB occurs mainly upstream of blocks (Altenhoff et al., 2008; Spensberger and Spengler, 2014), we define a separate target region for a CWB index (70°W-30°W/50°N-70°N) that is slightly equatorward and upstream of the target box used for the blocking index. This region corresponds to the largest CWB frequency when Greenland blocking occurs (see e.g., Michel and Rivière, 2011, and the Greenland blocking composites in Fig. 5d,e,f).

## 2.5 Anticyclone detection

We detect anticyclones using the method from Wernli and Schwierz (2006), which identifies the area covered by anticyclones using the outermost closed contour around a maximum in sea-level pressure. This procedure leads to problems over high topography, because the extrapolated sea-level pressure is very sensitive to near-surface temperatures. For this reason high topography is masked in many detection schemes for cyclones and anticyclones (c.f. intercomparsion in Neu et al., 2013).

As we are interested in anticyclones over Greenland, we thus adapt the procedure to use anomalies of 500-hPa geopotential with respect to the seasonal climatology as input to the anticyclone detection. Although about 200 hPa above Greenland's highest point, the 500-hPa level is the lowest level not intersecting the Greenland topography that is available for all models. We require a minimum height difference between the geopotential maximum and the outermost closed contour of 25 m (compared to 2 hPa in the original definition of the algorithm in Sprenger et al., 2017) and require a size of the anticyclone between 1 and $18 \, 10^6 \, \text{km}^2$ (consistent with the original definition in Sprenger et al., 2017).

We use this objective detection algorithm because even though a blocking can be considered as a stationary anticyclone, an anticyclone can occur without reversal of geopotential contours, which is the method used in this study to detect blocking. Thus, we are able to see if there are anticyclones over Greenland that are not linked to an overturning of a geopotential contour and without any minimum persistence.

### 2.6 Statistical significance

#### 2.6.1 For biases

The significance of biases is assessed with a two-sided t-test at a significance level of 90%. For the models, the 9-winter climatology is first computed for each member, then the ensemble mean and standard deviation are computed. Nine winters might be considered too short to accurately assess the blocking frequency due to its large interannual variability. However, using ERA-Interim, we show that none of the 30 climatologies of 9 consecutive winters (i.e. 1980-1989, 1981-1990, etc.) of blocking frequency is significantly different from the total 40-year (1979-2018) climatology (Fig. S2). To test the significance of biases, we compare the model mean to the observed 2006-2015 mean using an estimate of the variability from the standard deviation of 100 means of nine winters randomly chosen (with replacement) and non consecutive taken from the whole ERA-Interim period (1979-2018).

#### 2.6.2 For composites

The significance of the composites was performed using a bootstrap method. For each member, X random winter days are averaged together, X being the number of days corresponding to the number of blocked days (for Figs. 5 and 7) or to the number of days with CWB index above the 95th percentile (for Fig. 6). More precisely, following Brunner et al. (2017), we pick Y random days corresponding to the number of events in the considered member. These are the starting days of the events and the event duration is the same as in the member. For example, if we have two events lasting four consecutive days in the member, then two random days are picked along with their next three days (i.e. a total of four days per event). We take the ensemble mean and repeat this operation 1000 times. Finally, the percentile at which the composite value is located in the bootstrapped distribution is found and all grid points with percentiles below the 10th and above the 90th percentiles are considered as significant.

## 3 Models biases

This section documents the biases in the HAPPI models (the ensemble means) with respect to the ERA-Interim reanalysis. A comprehensive characterization of the atmospheric mean state bias in the HAPPI models was performed by Li et al. (2018), so we hereafter summarize the main results of that work relevant to the current study and complement them with an analysis of the biases in blocking and RWB frequencies.

### 3.1 Blocking bias

Like most CMIP5 models (Anstey et al., 2013; Dunn-Sigouin and Son, 2013; Masato et al., 2013), the HAPPI models generally have too few blocks over the North Atlantic during winter (Fig. 1, blue shading). In the North Atlantic sector, blocking occurs in a few preferred regions (Treidl et al., 1981; Dole and Gordon, 1983; Lupo and Smith, 1995). The maximum in the subtropics (Fig. 1f) is a manifestation of the semi-permanent Azores High rather than a high frequency of blocking events (Davini et al.,

2014). A second blocking region (∼ 8-9%) is found over north-western Europe and a third over Greenland (∼ 5-6%, black box). All models underestimate blocking in the North Atlantic sector, with some models (e.g. CAM4-2degree) showing almost no blocking at all (i.e. a negative bias as large in magnitude as the climatology). All models exhibit significant (non-dotted) negative biases over Greenland and UK, with MIROC5 having also a significant positive bias southwest of Greenland. MIROC5 is the model with the smallest bias and ECHAM6.3-LR the model with the second lowest bias over Greenland. This is also obvious from Table 1 as ECHAM6.3-LR and MIROC5 are the models with the highest ensemble mean blocking frequencies and the only models where blocking occurs in all ensemble members.

Accurate Greenland blocking can occasionally be reproduced by a few members of some models even though these models exhibit negative biases in the ensemble mean. This highlights the advantage of using a large number of ensemble members (or long simulations) to sample relatively rare events such as blocking. Figure 2 shows the distributions of the nine-winter mean frequencies of Greenland blocking for each model (colored bars) and the lowest and highest blocking frequencies (dashed vertical lines for 5.6 and 14.1%) from the 31 mean DJF frequencies obtained for every possible decade (1979-1988, 1980-1989, etc) covering the ERA-Interim period (1979-2018). Three models of the HAPPI ensemble, CanAM4, NorESM1-HAPPI, and CAM4-2degree, have much lower blocking frequencies than ERA-Interim and only 9%, 6% and 2% of their distributions fall within ERA-Interim's range. These three models can on occasion simulate blocking with a frequency close to ERA-Interim's but they seem to lack an ingredient for blocking formation that can systematically increase the total blocking frequency in every member. Remarkably, more than half of CAM4-2degree's members have no blocking over Greenland (gray bar in Fig. 2a). ECHAM6.3-LR and MIROC5 perform better, with a fair number of members able to simulate ERA-Interim's blocking frequency for the decade 2006-2015 (12.68%, represented by the black solid line in Fig. 2). 74% of MIROC5 members and 47% of ECHAM6.3-LR members are within the full range of ERA-Interim (5.6-14.1%), with MIROC5's distribution overshooting ERA-Interim's and ECHAM6.3-LR's distribution undershooting ERA-Interim's. Our result for MIROC5 is in agreement with Masato et al. (2013) who showed that the CMIP5 coupled version of MIROC5 has a tendency to overestimate the GB frequency and to shift it over the Labrador Sea. Overall, MIROC5 is the model with the closest ensemble mean GB frequency to ERA-Interim.

## 3.2 Large-scale atmospheric circulation biases

Similar to the CMIP5 ensemble mean, the majority of the HAPPI models exhibit a too zonal and too strong North Atlantic eddy-driven jet in winter (as illustrated by the positive bias in the low-level zonal wind in Fig. S3), with the exception of MIROC5 whose eddy-driven jet is too weak. ECHAM6.3-LR best reproduces the DJF climatological low-level zonal winds. All models underestimate the southwest-northeast tilt of the North Atlantic low-level jet, with ECHAM6.3-LR and CanAM4 having the most realistic North Atlantic tilt (not shown).

As blocking is detected from Z500, any bias in the mean state and variability of this field can influence the representation of blocking. The mean state bias is characterized by a trough that is not deep enough over eastern North America (60°W) and a ridge not pronounced enough over western Europe in most models (Fig. S4). This is in accordance with the biases in stationary waves, defined by the 500-hPa geopotential deviation from the zonal mean, exhibiting a weakened ridge consistent with the

too zonal climatological jet, in four out of the five models (Fig. S5). MIROC5's Z500 mean state bias exhibits a meridional dipole of opposite sign compared to the other models with a positive bias north of 50°N and a negative bias south of 50°N, respectively (Fig. S4d). ECHAM6.3-LR is also slightly different and shows only a slight negative bias close to Newfoundland at 50°N (Fig. S4c). This means that the trough at 60°W is too pronounced in ECHAM6.3-LR and not pronounced enough in MIROC5 in association with a too strong and too weak meridional gradient of Z500, respectively. MIROC5 is the model with the widest ridge, which extends too much to the west (Fig. S5d).

Biases in the mean state of the atmosphere could result from biases in the simulated variability (e.g. Kidston and Gerber, 2010; Kwon et al., 2018). For example, if Greenland blocking is too frequent with the jet too often shifted southward, we expect a southern bias in the mean wind state. Here, we examine the zonal wind variability by computing the standard deviation of the daily zonal wind at 850 hPa for each ensemble member separately before averaging over all members (Fig. 3). Similar results are observed for the wind at 250 hPa (not shown). In the reanalysis, the highest variability of the daily zonal wind (i.e. the highest standard deviations) in the North Atlantic is co-located with the climatological jet stream end and extends eastwards of 60°W over a broad latitudinal range (∼ 40°-70°N, Fig. 3). All HAPPI models exhibit standard deviation values similar to ERA-Interim, however, only on the poleward side of the climatological jet between the southern tip of Greenland and Iceland. This suggests that the simulated North Atlantic daily jet is too infrequently in a southward-shifted position, similar to the results found in Kwon et al. (2018). MIROC5 and ECHAM6.3-LR are the models with the largest variability on the equatorward side of the mean jet (30°N, Fig. S6) hence the smallest bias in wind variability.

### 3.3 Rossby wave breaking bias

RWB has been shown to play an important role for blocking and the formation and maintenance of weather regimes (e.g. Swenson and Straus, 2017). The ERA-Interim climatology of RWB frequency shows that AWB is most frequent on the equatorward side of the mean jet (compare red contours to gray shading in Fig. S7f) while CWB is less frequent than AWB but shows a maximum frequency on the poleward side of the mean jet (compare blue contours to gray shading in Fig. S7f) (see also Martius et al., 2007). However, both types of RWB are generally more frequent than blocking. Since blocking formation often involves RWB (Altenhoff et al., 2008; Michel and Rivière, 2011; Masato et al., 2012; Spensberger and Spengler, 2014; Woollings et al., 2018), it is important to know how climate models represent RWB.

Most HAPPI models show a similar RWB pattern as ERA-Interim (Fig. S7), with the largest frequencies over the ocean, but their absolute values are generally too low (negative bias with blue shading in Fig. 4). Such negative biases in both AWB and CWB were also found for previous models versions (e.g., ECHAM5-HAM T63 in Béguin et al., 2013) using a different approach to detect wave breaking. MIROC5 stands out with in general too little AWB where ERA-Interim has a frequency maximum (blue shading superimposed to the grey contours in Fig. 4d right) and too much AWB to the north of this maximum (red shading in Fig. 4d left). MIROC5 is the model with the strongest negative biases in CWB (blue shading in Fig. 4 right). Overall, ECHAM6.3-LR is the model exhibiting the smallest biases in both AWB and CWB. The bias in CWB is also obvious in Table 1 with MIROC5 having the weakest mean CWB index and ECHAM6.3-LR the largest.

**Table 1.** For all models and experiments, this table provides the number of members which have at least one day with Greenland blocking, as defined by the 10% threshold of the blocking index, out of the total number of members, the mean wintertime frequency of blocked days (blocking frequency in the Table) over those selected members, and the ensemble mean wintertime frequency of blocked days taking into account all members (as in Fig. 2). If all members exhibit blocked days, the mean frequency (4th column) equals the full ensemble mean frequency (5th column). The last column gives the (ensemble) wintertime mean of the CWB index (all winter days are taken into account) in % of the box area as defined in Section 2.4. A value of 100% would mean that every grid point in the box exhibits CWB.

| Model/Reanalysis | Experiment | #members | Blocking frequency (blocking members) | Blocking frequency (all members) | (Ens.) mean CWB index |
|---|---|---|---|---|---|
| CAM4-2degree | Present | 213/501 | 1.75% | 0.74% | 7.6% |
| CanAM4 | Present | 97/100 | 2.88% | 2.79% | 10.1% |
| ECHAM6.3-LR | Present | 100/100 | 5.69% | 5.69% | 11.6% |
| MIROC5 | Present | 100/100 | 12.47% | 12.47% | 4.7% |
| NorESM1-HAPPI | Present | 119/125 | 2.54% | 2.42% | 8.4% |
| CAM4-2degree | Future | 199/501 | 1.48% | 0.59% | 7.4% |
| CanAM4 | Future | 95/100 | 2.97% | 2.82% | 10.4% |
| ECHAM6.3-LR | Future | 100/100 | 4.76% | 4.76% | 11.5% |
| MIROC5 | Future | 100/100 | 9.90% | 9.90% | 4.7% |
| NorESM1-HAPPI | Future | 121/125 | 2.21% | 2.14% | 8.2% |
| ERA-Interim | 2006-2015 | 1/1 | 12.68% | - | 11.1% |

## 4 Dynamics of Greenland blocking

As seen in the above description of the bias in the HAPPI models, ECHAM6.3-LR and MIROC5 are noticeably different from the other three models. These two models best reproduce the Greenland blocking climatology seen in ERA-Interim despite
contrasting biases in the atmospheric mean state (Z500, U850, RWB) and variability (Z500, U850) over the North Atlantic. The models' differences are most obvious southwest of Greenland where MIROC5 shows positive bias in AWB frequency, Z500, stationary wave and a negative bias in CWB frequency and U850 while ECHAM6.3-LR shows the opposite bias sign or negligible bias. Table 1 summarizes the different behaviour of MIROC5 and ECHAM6.3-LR: MIROC5 has the largest mean blocking frequency and weakest mean CWB index whereas it is the opposite for ECHAM6.3-LR. In the following, we will
focus on these two models and compare the mechanisms leading to Greenland blocking.

### 4.1 Composites over blocked days

In agreement with ERA-Interim, ECHAM6.3-LR and MIROC5 exhibit an anticyclonic anomaly over Greenland and stronger westerly zonal wind to the south of the North Atlantic during blocked days (Fig. 5). However, MIROC5 does not exhibit an enhanced CWB frequency south of Greenland, as seen in ECHAM6.3-LR and ERA-Interim (compare the composites in Fig.

5e with panels d and f). This is curious, as several studies have shown that one of the key drivers of Greenland blocking is an enhanced frequency of CWB (Woollings et al., 2008; Michel and Rivière, 2011; Swenson and Straus, 2017; Madonna et al., 2019), which, through convergence of meridional eddy momentum fluxes, acts to shift the jet equatorwards (Thorncroft et al., 1993; Rivière and Orlanski, 2007). Table 1 shows the number of members in each ensemble used in the composites over the blocked days. The zonal wind at 850 hPa is anomalously south and zonal from North America to the Mediterranean for both models and ERA-Interim (Fig. 5j-l). Since the method detecting geopotential contours reversal is used to identify blocking, all composites exhibit a pronounced ridge over Greenland with a cyclonic overturning over the Labrador Sea. However, the associated anticyclonic (positive) anomaly of geopotential is larger for ECHAM6.3-LR and ERA-Interim than for MIROC5 (Fig. 5a-c). Even though MIROC5 does not exhibit enhanced CWB south of Greenland compared to ECHAM6.3-LR and ERA-Interim (Fig. 5d-f), the three of them show a slight positive anomaly of AWB frequency close to Iceland hinting at an $\Omega$-shape of the blocking (Fig. 5g-i) however smoothed in the composite of geopotential height. In essence, the comparison between ERA-Interim, ECHAM6.3-LR and MIROC5 demonstrates that MIROC5 produces a realistic blocking frequency but for unclear reasons.

## 4.2 Discussion

Of the five models examined here, ECHAM6.3-LR is the least biased in terms of mean state, variability, and RWB, and the Greenland blocking frequency is only underestimated by 2-3% as seen on Fig. 1c. Only MIROC5 has more realistic Greenland blocking, although, as shown previously, it shows much larger biases in the other fields. In this section, we discuss the RWB biases, how CWB modifies the atmospheric circulation, and explore potential reasons explaining the above results.

RWB can drive the eddy-driven jet position by accelerating/decelerating the wind in specific locations but the link between RWB biases and wind biases is not so simple. However, we note that models with a too strong zonal wind over northern Europe (CAM4-2degree, CanAM4, ECHAM6.3-LR and NorESM1-HAPPI in Figs. S3 and S8) are associated with a positive bias in AWB over southern Europe (Fig. 4): there are too many AWB events forcing the jet too far northwards. ECHAM6.3-LR and ERA-Interim exhibit an anticyclonic reversal of the absolute vorticity isolines south of the jet ($\sim$30°N) linked to the meridionally-confined maximum of AWB frequency. In contrast, for MIROC5, the meridionally wide area of AWB reflects a smooth isoline reversal in the absolute vorticity field (Fig. S9d). Moreover, the meridional gradient of absolute vorticity over the North Atlantic in MIROC5 is clearly very weak compared to ECHAM6.3-LR and ERA-Interim, especially over the western side of the ocean basin, because of the weak mean zonal wind and its meridional gradient (absolute vorticity depends on the vertical component of the wind curl). This negative bias in absolute vorticity in addition to the weak trough in the stationary wave pattern over the Labrador Sea could explain the absence of CWB in MIROC5 (Figs. S9d and S5d, respectively, Barnes and Polvani, 2013) whereas some waves can propagate and break anticyclonically anywhere in the North Atlantic (Figs. 4d and S7). Barnes and Hartmann (2011) found that a weak absolute vorticity gradient poleward of the jet inhibits CWB occurrence. Although it hampers CWB, the weak absolute vorticity gradient may also promote blocking formation if we assume that potential vorticity behaves similarly to the absolute vorticity. Luo et al. (2019) showed in an idealised set-up that, at high latitudes, a weak mean meridional gradient of potential vorticity, associated with weak mean wind, leads to reduced

**Table 2.** Ensemble mean and spread (standard deviation over the members) of the number of days in each category for ECHAM6.3-LR and MIROC5 and total number of days in each category for ERA-Interim. Unit: days. The CWB/no CWB categories distinguish the days for which the spatially averaged CWB frequency, as defined in Section 2.4, is greater than 0 / equals 0. GB stands for Greenland Blocking. The GB/no GB categories distinguish the blocked days from the non-blocked days, as defined in Section 2.3. The sum of the number of days in the four categories for each model and ERA-Interim equals the total number of winter (DJF) days in the decade 2006-2015.

|       | ECHAM6.3-LR | | MIROC5 | | ERA-Interim | |
|-------|-------------|---------------|-------------|---------------|------|--------|
|       | CWB | no CWB | CWB | no CWB | CWB | no CWB |
| GB    | $36.2 \pm 14.9$ | $10.0 \pm 6.0$ | $51.2 \pm 13.6$ | $49.8 \pm 15.2$ | 77 | 26 |
| no GB | $431.9 \pm 20.5$ | $333.9 \pm 22.6$ | $323.4 \pm 20.1$ | $385.6 \pm 24.4$ | 406 | 303 |

energy dispersion, enhanced nonlinearity, and more persistent eddy forcing, favouring long and intense blocking. Even though MIROC5 does not exhibit more intense or longer blocking than ECHAM6.3-LR, this mechanism could also trigger blocking thus enhancing its frequency.

Our results suggest that Greenland blocking in MIROC5 is not necessarily linked to CWB, but that CWB can nevertheless lead to a ridge over Greenland and a local enhancement of the zonal wind. Figure 6 shows composites of the days with a CWB index (defined in Sec. 2.4) larger than the 95th percentile for ECHAM6.3-LR, MIROC5, and ERA-Interim. We see that when there is CWB southwest of Greenland, there is a positive geopotential anomaly (Fig. 6a-c), which is only sometimes associated with blocking (Fig. 6m-o). This could be due to the fact that not all CWB events trigger blocking and/or that CWB events mainly occur during blocking formation but are much less frequent during the mature stage of blocks. If we account for some time for the block to form, we observe a slight increase in blocking frequency 1-2 days after CWB occurrence (not shown). The same is true for ERA-Interim, therefore, the absence of CWB during Greenland blocking in MIROC5 (Fig. 5e) is not due to a timing issue. MIROC5 exhibits more frequent blocking events with only a slightly longer duration (Fig. S10). Thus, the high Greenland blocking frequency in MIROC5 results mainly from more blocking events detected rather than a longer duration of these events.

Table 2 shows that Greenland blocking occurs as frequently with CWB (GB-CWB) as without CWB (GB-no CWB) for MIROC5 (51.2 versus 49.8 days), whereas for ECHAM6.3-LR and ERA-Interim Greenland blocking occurs most frequently with CWB (36.2 versus 10.0 days for ECHAM6.3-LR and 77 versus 26 days for ERA-Interim). This difference probably arises from the lack of CWB in MIROC5. Also, the composites of the 500-hPa geopotential for the category GB-no CWB exhibits a westward shift of the anticyclonic anomaly compared to the GB-CWB category (see first row in Figs. S11, S12 and S13). This may reflect the blocking at a later stage of its lifetime as recently shown by Drouard et al. (2021) for the blocking of cyclonic type typical over Greenland. Whether or not CWB occurs during Greenland blocking, the low-level zonal wind is always stronger south over the North Atlantic (see columns (a) and (b) of the fourth row in Figs. S11, S12 and S13).

Both ECHAM6.3-LR and MIROC5 tend to overestimate the presence of anticyclones, defined in Sec. 2.5, over Greenland (Fig. S14). It seems that, for MIROC5, a weak mean zonal wind associated with the biases in geopotential and absolute

vorticity favours the presence of anticyclones (positive geopotential height anomalies) over Greenland, whether or not CWB occurs. To conclude, while in reanalysis and ECHAM6.3-LR, CWB seems an important ingredient for Greenland blocking, this mechanism is not equally present in MIROC5.

## 5 Future changes in Greenland blocking and RWB

After having analyzed the dynamics of GB in the HAPPI large ensemble, we are interested to see how future changes in blocking are linked to changes in its driver, namely CWB, in ECHAM6.3-LR where CWB are fairly simulated, and in MIROC5, the model with the best Greenland blocking frequency compared to ERA-Interim.

In agreement with previous studies using CMIP3, CMIP5, and CMIP6 experiments (e.g., Sillmann and Croci-Maspoli, 2009; Barnes et al., 2012; Masato et al., 2013; Woollings et al., 2018; Davini and D'Andrea, 2020), we note a weak and non-significant decrease between the present and future experiments in the percentage of blocked days (see Table 1) and in the ensemble mean blocking frequency over Greenland, in particular for ECHAM6.3-LR (up to -0.5%, Fig. S15c) and MIROC5 (up to -1.5%, Fig. S15d). This decrease is weaker compared to the studies cited above (e.g., -2 to -4% over Greenland in the CMIP multi-model mean responses in Fig. 6a-c of Davini and D'Andrea, 2020) mainly because the HAPPI future experiments represent a very mitigated warming scenario with a global mean temperature increase of $+2°C$ relative to pre-industrial climate compared to the $+3.2$ to $5.4°C$ at the end of the 21st century for the Representative Concentration Pathway 8.5 of CMIP5 (IPCC, 2013). Previous studies showed that the decrease in Greenland blocking frequency seems linked to the poleward shift of the North Atlantic eddy-driven jet, as expected from the response to changes in baroclinicity, mainly at upper levels, due to global warming (Harvey et al., 2014; Shaw et al., 2016; Yin, 2005). However, even though some studies found decreasing trends in blocking frequencies (Sillmann and Croci-Maspoli, 2009; Masato et al., 2013; Matsueda and Endo, 2017; Woollings et al., 2018), such trends are often found to not be significant and to be very dependent on the metric and field used to detect blocking (Collins et al., 2019; Wachowicz et al., 2020). The composites over the blocked days for the future experiment are very similar to the composites for the present period (compare Fig. 7 to the left and middle columns of Fig. 5). The blocking index used in the present study is not affected by the increase in geopotential height due to global warming (Christidis and Stott, 2015) contrary to other Greenland blocking indices (Wachowicz et al., 2020).

Although ECHAM6.3-LR and MIROC5 predict decreased Greenland blocking, there is no obvious decrease in CWB or increase in AWB as it would be expected from previous studies. Global warming is expected to enhance the upper-tropospheric baroclinicity (Harvey et al., 2014), which affects the nature of breaking of Rossby waves, leading to more AWB in an idealized zonally symmetric quasi-geostrophic model (Rivière, 2011). Barnes and Hartmann (2010) and Barnes and Polvani (2013) related the future decrease in blocking frequency to a northward shifted jet that hinders CWB on the poleward flank of the jet over the North Atlantic. In the very mitigated scenario of the HAPPI models, AWB become less frequent at almost all longitudes around $30°N$ over the oceanic basins of the Northern Hemisphere in winter (red dashed contours in Fig. S16). Over the North Atlantic, the CWB frequency does not change (noisy field with amplitudes below first contour level) and AWB are less frequent for both ECHAM6.3-LR and MIROC5 despite the 850-hPa zonal wind responses being different (Fig. 8). For

ECHAM6.3-LR, the zonal wind is accelerated where AWB is less frequent, west of 20°W and is accelerated between 50°N-60°N in relation with more AWB to its southeastern side (Fig. 8a). For MIROC5, the link between the zonal wind and RWB responses is not clear as the zonal wind is accelerated to the north at $\sim 60°$N between 80°W-10°E despite the decrease in AWB especially over the western part of the oceanic basin (west of 30°W) (Fig. 8b). Therefore, in these two HAPPI models, the link between the changes in the Greenland blocking frequency and its driver is not obvious nor as expected from previous studies.

## 6 Conclusions

In this study, we examine the representation of Greenland blocking in large ensembles of climate models simulations as well as the role of CWB as a driver. As blocking is a relatively rare event ($\simeq$10-20% of the time in winter), large ensembles are required to ensure a sufficient number of events to be able to draw robust conclusions. In line with previous studies which analysed various climate models (e.g. the CMIP5 multi-model ensemble, Anstey et al., 2013; Dunn-Sigouin and Son, 2013), we find that Greenland blocking frequency is strongly underestimated in three out of the five HAPPI models used here. We see that the underestimation of GB frequency is linked to too little variability in the low-level zonal wind over the southern part of the North Atlantic, on the equatorward flank of the eddy-driven jet. This lack of variability is also apparent in the negative bias in CWB, the main driver of Greenland blocking identified in reanalyses, which acts to push the eddy-driven jet to the south and advect low potential vorticity air poleward. We focus on the two models that have a fair representation of Greenland blocking frequency: ECHAM6.3-LR exhibits the smallest bias in the mean state and only slightly underestimates Greenland blocking frequency for the reasons cited above (low variability in wind to the south and CWB not frequent enough), while MIROC5 has large biases in mean climate but is best at representing Greenland blocking frequency. MIROC5 produces more events, which on average last slightly longer than in the other models. However, the mechanisms leading to blocking in MIROC5 appear to be different to those in ECHAM6.3-LR and documented for reanalyses. This difference is most apparent in CWB occurrence, which is severely underestimated, and thus at odds with the accurate Greenland blocking frequency.

Rossby wave breaking patterns are quite well represented by most models, MIROC5 being the exception, but there is still a negative bias for both AWB and CWB almost everywhere in the European-North Atlantic domain and a positive bias of AWB over the Mediterranean. The link between RWB and Greenland blocking in ECHAM6.3-LR is similar to ERA-Interim with large CWB frequency during GB events and some blocking events when CWB occur southwest of Greenland. However, the link between CWB and Greenland blocking in MIROC5 is not clear. Indeed, MIROC5 exhibits a strong negative bias in CWB over most of the Northern Hemisphere. Even though there is a reversal of the isohypses (lines of equal geopotential), the CWB frequency and the associated geopotential anomaly are very weak during blocking events but we show that MIROC5 can still produce blocking from CWB events. Therefore, the dynamical link between CWB and Greenland blocking is present but not the main ingredient in triggering Greenland blocking in MIROC5. There must then be another process in this model that favors the northwards advection of airmasses over Greenland.

In agreement with previous studies, ECHAM6.3-LR and MIROC5 both exhibit a decreased frequency of Greenland blocking in the future experiments. However, we find that the decrease is not significant and not clearly linked to a reduced frequency of

CWB, as could have been expected from previous studies (e.g., Barnes and Hartmann, 2012). Moreover, Greenland blocking composites of the geopotential, zonal wind, and RWB for the future period are very similar to the composites for the present period.

Our study highlights that, in order to evaluate blocking representation in climate models, we should not just consider biases in the mean state. It is also important to evaluate the representation of the known mechanisms that lead to blocking, such as CWB, which is an indicator for the eddy-mean flow interaction. Davini et al. (2017) started to tackle this issue by studying the representation of eddies in one climate model with various spatial resolutions, finding that higher resolution simulations do not necessarily better represent eddies. A better understanding of the biases sources in the mechanisms leading to blocking in climate models is crucial to reduce those biases and improve the prediction of future changes.

*Code and data availability.* The method to identify blocking is described in Scherrer et al. (2006). The RWB and anticyclone detection algorithms can be found at https://doi.org/10.5281/zenodo.4639624 (Spensberger, 2021) and https://git.app.uib.no/Clemens.Spensberger/ dynlib. The HAPPI dataset is available at https://portal.nersc.gov/c20c/data.html. The ECMWF ERA-Interim reanalysis is available at https: //apps.ecmwf.int/datasets/data/interim-full-daily/levtype=pl/ (Dee et al., 2011).

*Author contributions.* C. Michel and E. Madonna equally contributed to the study (analysis and writing). C. Spensberger contributed to the writing and performed the anticyclone detection analysis. C. Li and S. Outten contributed to the manuscript with comments and suggestions.

*Competing interests.* Camille Li is a member of the editorial board of the journal.

*Acknowledgements.* The authors are grateful to the three anonymous reviewers whose comments helped to improve the manuscript. This work was partly funded by the Bjerknes Centre for Climate Research in Bergen through the Fast Track Initiative granted to Clio Michel and the strategic project EMULATE to Erica Madonna. The authors thank the European Centre for Medium-range Weather Forecast for providing the ERA-Interim analyses and the HAPPI project for providing the very large ensembles of simulations. We are also grateful to ETH Zurich for sharing their blocking and anticyclone detection algorithms. We acknowledge the Norwegian e-infrastructure for research and education UNINETT Sigma2 (project NS9770K) and the Research Council of Norway (projects 295046 KeyCLIM and 275269 COLUMBIA). This research used science gateway resources of the National Energy Research Scientific Computing Center, a DOE Office of Science User Facility supported by the Office of Science of the U.S. Department of Energy under Contract No. DE-AC02-05CH11231.

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

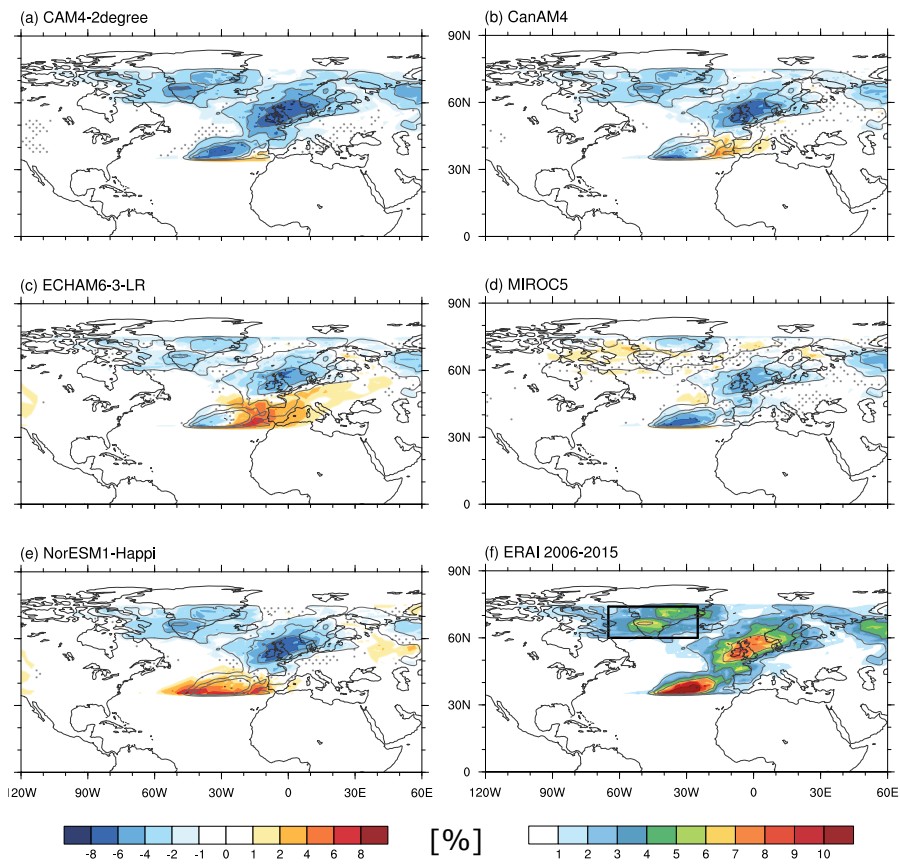

**Figure 1.** (a-e) Bias in winter (DJF) blocking frequency for the five models (ensemble mean of the blocking frequency minus ERA-Interim) and (f) ERA-Interim DJF blocking climatology for 2006-2015 (in frequency, as %). Dark gray lines show the smoothed 2, 4 and 6% contours for ERA-Interim (2006-2015). The black box shows the main region of Greenland blocking in ERA-Interim. Biases that are not significant at the 90% level are dotted and there are no dots where there is no blocking.

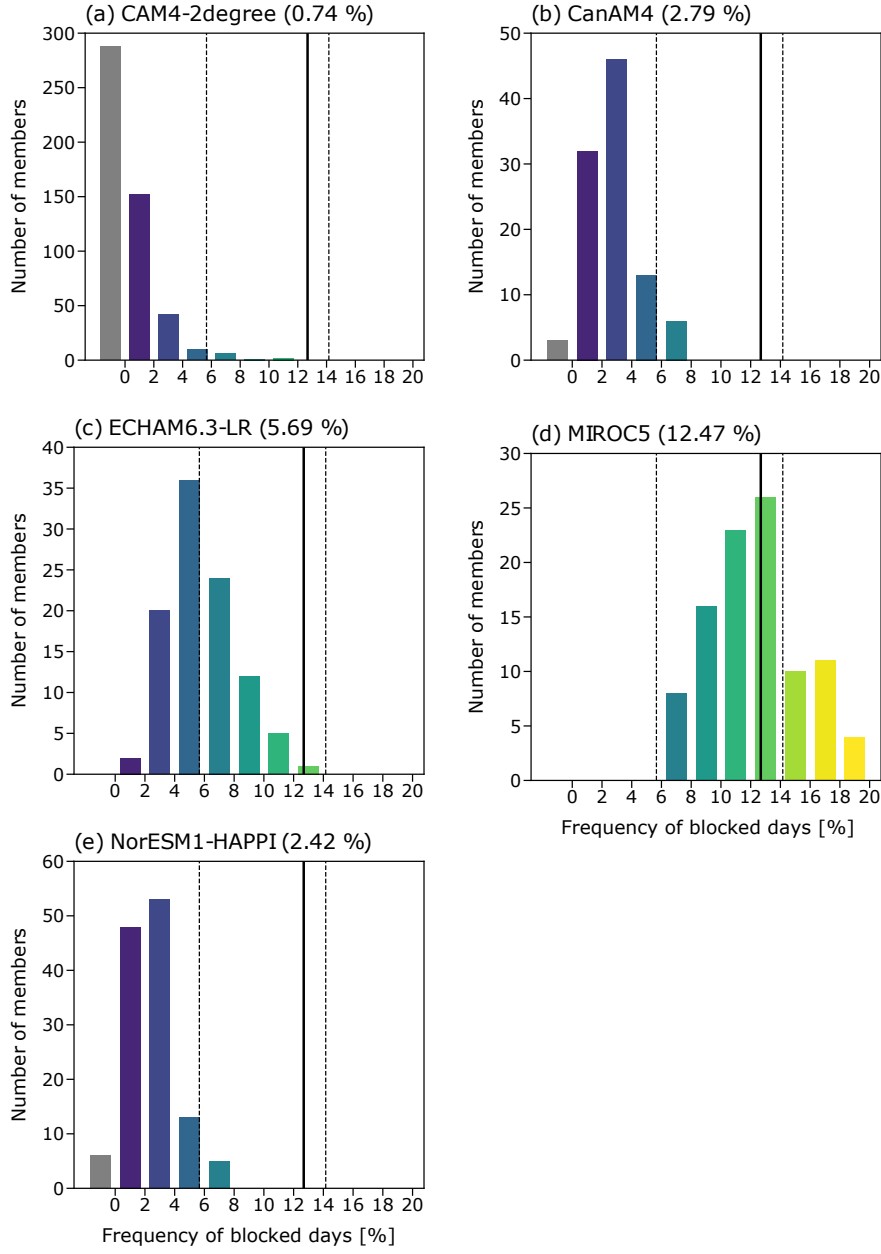

**Figure 2.** Distributions of the nine-winter (DJF) mean frequency of Greenland blocking days (in 2% bins) for each ensemble. The mean frequency of each model is shown in the title and given in Table 1. Shown in every panel is the mean frequency of Greenland blocking days from ERA-Interim for 2006-2015, which is 12.68% (black line), and the minimum/maximum frequencies of blocking days from nine-consecutive winters for the whole ERA-Interim period of 1979-2018, which are 5.66 and 14.16% respectively (dashed lines). Gray bars show the number of members with no GB blocking in the nine-year period.

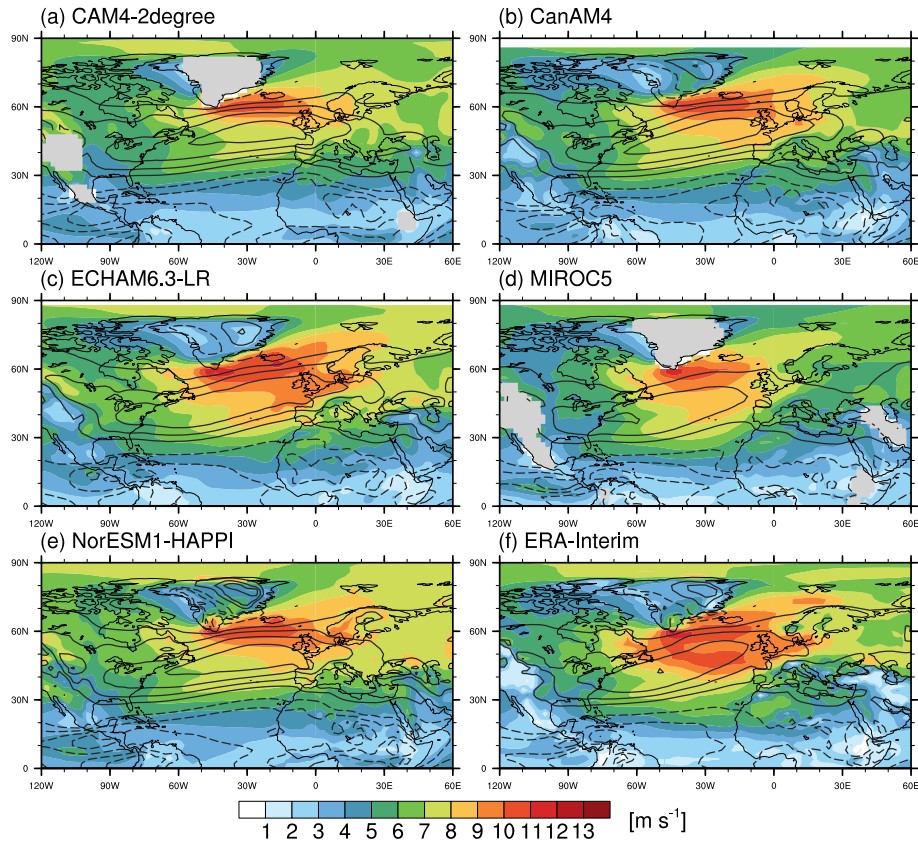

**Figure 3.** Ensemble means of the DJF mean daily standard deviation of the 850-hPa zonal wind (shading, in m s$^{-1}$) and of the DJF 850-hPa zonal wind climatology (contours, interval: 3 m s$^{-1}$, zero contour omitted, negative values with dashed lines) for (a) CAM4-2degree, (b) CanAM4, (c) ECHAM6.3-LR, (d) MIROC5, and (e) NorESM1-HAPPI. The daily standard deviation is calculated for each member separately and then averaged over the ensemble. (f) DJF mean daily standard deviation (shading, in m s$^{-1}$) and climatology (contours, interval: 3 m s$^{-1}$, zero contour omitted, negative values with dashed lines) of the 850-hPa zonal wind for the period 2006-2015 of ERA-Interim.

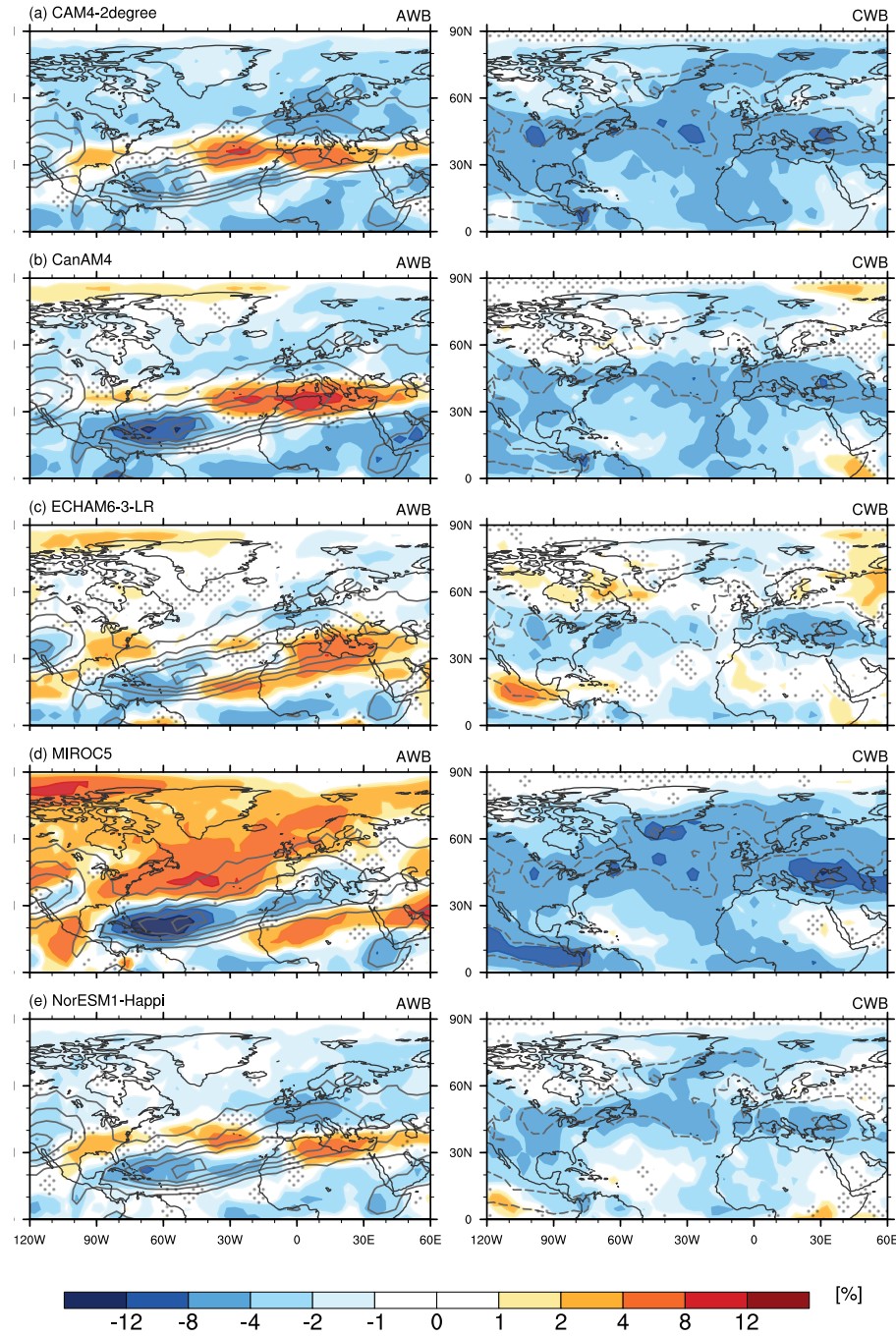

**Figure 4.** Bias in anticyclonic (AWB, left) and cyclonic (CWB, right) wave breaking for the five HAPPI models. Bias as shown as frequencies (in % of time), while the ERA-Interim climatology for the period 2006-2015 is shown in contours (starting at 10%, in 5% steps, left solid for AWB and right dashed for CWB). Black dots mark biases that are not statistically significant.

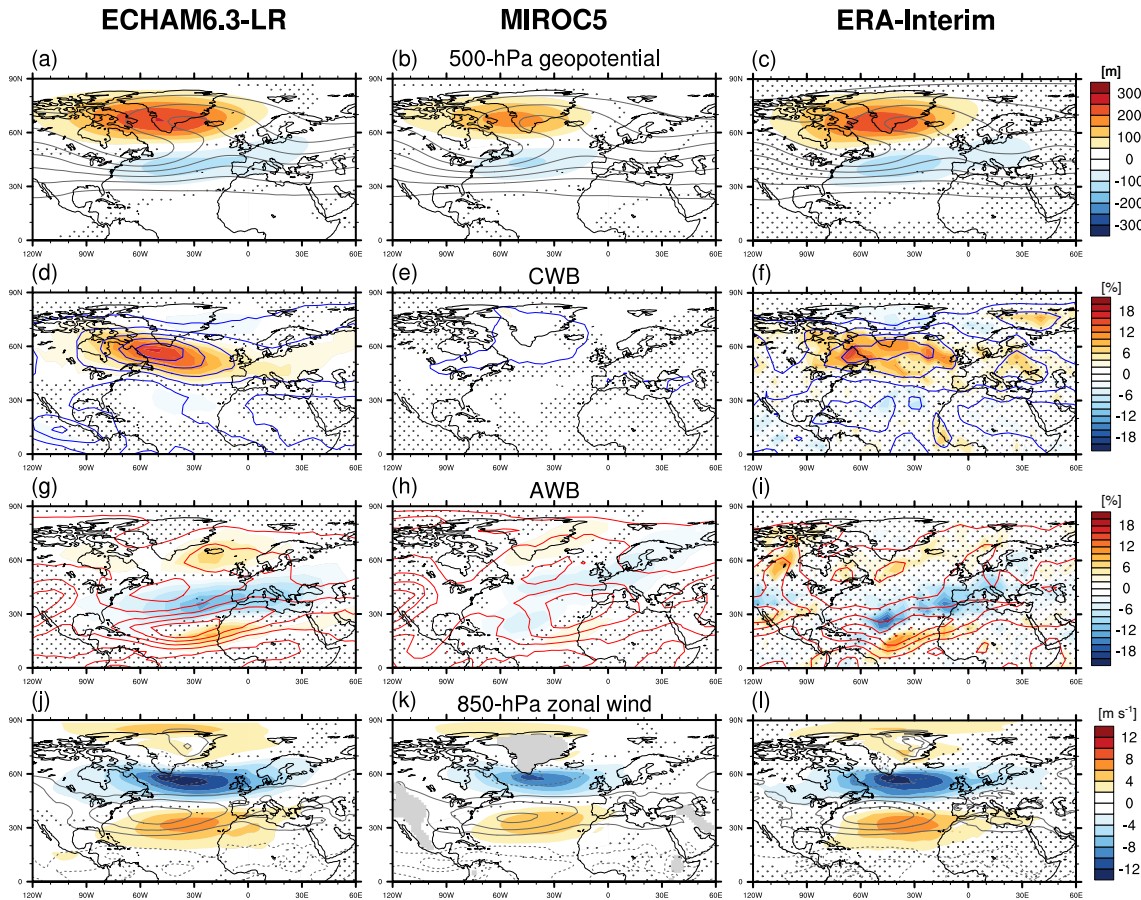

**Figure 5.** Ensemble means of the circulation anomalies during GB days for (left) ECHAM6.3-LR, (centre) MIROC5, and (right) ERA-Interim. (a,b,c) 500-hPa geopotential (contours are drawn every 100 m from 5000 to 6000 m) and anomalies (shading, in m). (d,e,f) Cyclonic wave breaking frequency (first contour and interval: 5%) and anomalies (shading, in %). (g,h,i) Same as (d,e,f) but for anticyclonic wave breaking frequency. (j,k,l) Zonal wind at 850 (first contour and interval: 4 m s$^{-1}$, zero contour omitted and dashed contours for negative values) and anomalies (shading, in m s$^{-1}$). Anomalies are deviations from the 10-year DJF climatology and only members with at least one blocked day are used for the composites. Dotted areas are not significant at the 10% level with the significance calculated using a bootstrap method. For ERA-Interim, the AWB and CWB frequency fields (contours) have been smoothed for better readability.

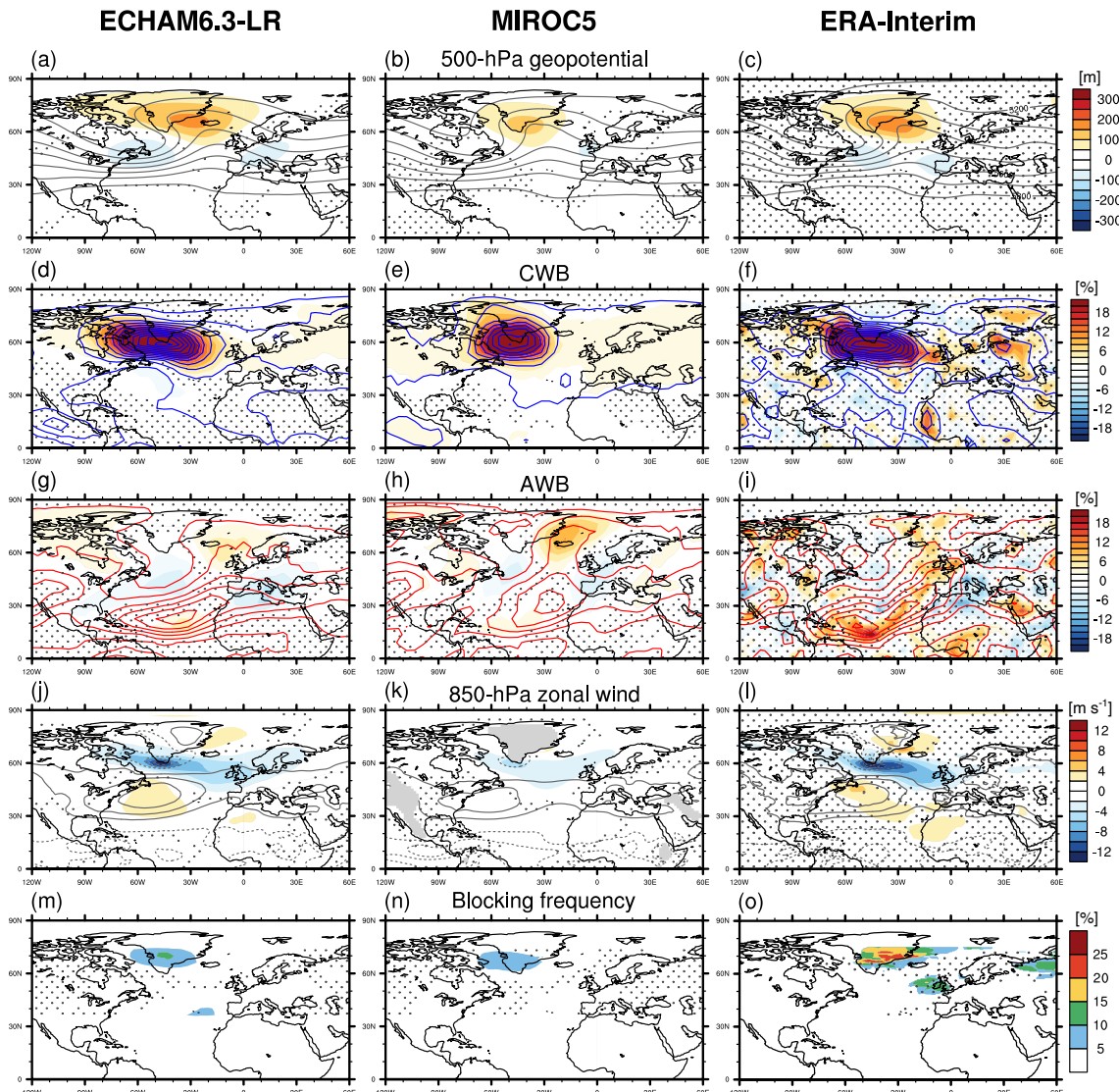

**Figure 6.** Ensemble means of the circulation anomalies during CWB days for (left) ECHAM6.3-LR, (centre) MIROC5, and (right) ERA-Interim. (a,b,c) 500-hPa geopotential (contours are drawn every 100 m from 5000 to 6000 m) and anomalies (shading, in m). (d,e,f) Cyclonic wave breaking frequency (first contour and interval: 5%) and anomalies (shading, in %). (g,h,i) Same as (d,e,f) but for anticyclonic wave breaking frequency. (j,k,l) Zonal wind at 850 hPa (first contour and interval: 4 m s$^{-1}$, zero contour omitted and dashed contours for negative values) and anomalies (shading, in m s$^{-1}$). (m,n,o) Blocking frequency (unit: fraction of the time with CWB). Dotted areas are not significant at the 10% level with the significance calculated using a bootstrap method. For ERA-Interim, the AWB and CWB frequency fields (contours) have been smoothed for better readability.

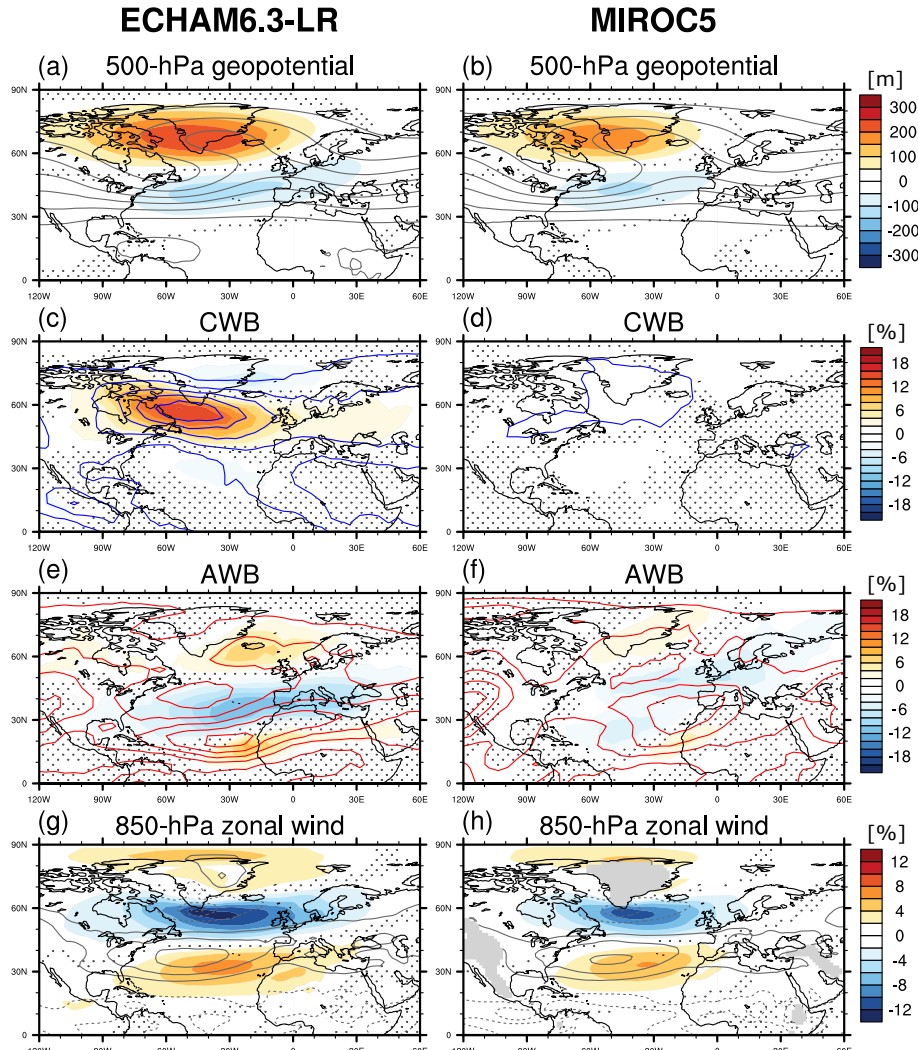

**Figure 7.** Ensemble means of the circulation anomalies during Greenland blocking days for (left) ECHAM6.3-LR and (right) MIROC5 future experiments. (a,b) 500-hPa geopotential (contours are drawn every 100 m from 5000 to 6000 m) and anomalies (shading, in m). (c,d) Cyclonic wave breaking frequency (first contour and interval: 5%) and anomalies (shading, in %). (e,f) Same as (c,d) but for anticyclonic wave breaking frequency. (g,h) Zonal wind at 850 hPa (first contour and interval: 4 m s$^{-1}$, zero contour omitted, dashed contours for negative values) and anomalies (shading, in m s$^{-1}$). Anomalies are deviations from the 10-year DJF climatology and only members with at least one blocked day are used for the composites. Dotted areas are not significant at the 10% level with the significance calculated using a bootstrap method.

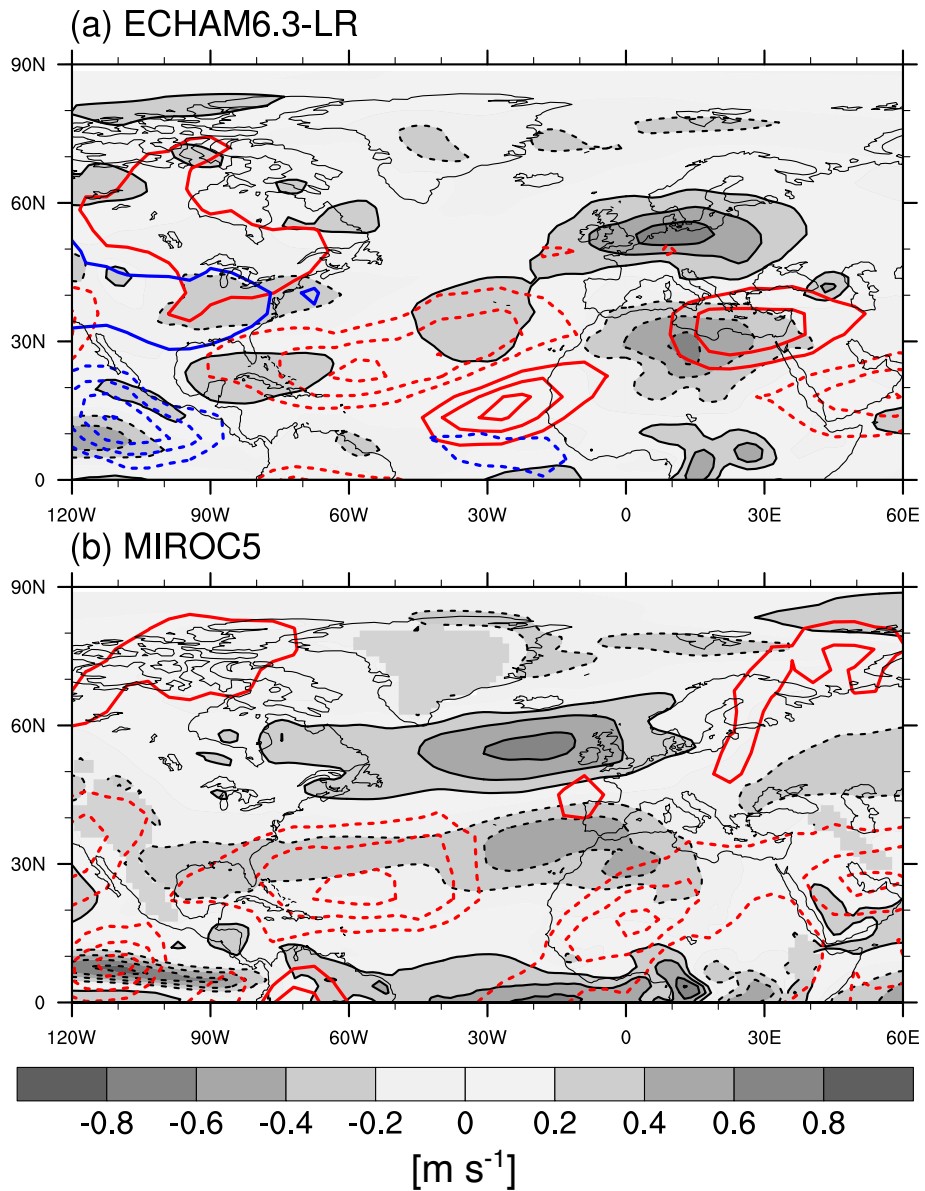

**Figure 8.** DJF ensemble mean responses (Future minus Present) of the Rossby wave frequency and 850-hPa zonal wind for (a) ECHAM6.3-LR and (b) MIROC5. Blue (red) contours show the responses for the (anti)cyclonic wave breaking frequency (first contour and interval: 0.005 day$^{-1}$). The gray shading and black contours show the response of the 850-hPa zonal wind (in m s$^{-1}$). The zero contours are omitted and dashed contours represent negative values.