# Peer review of "Dynamical drivers of Greenland blocking in climate models"

_Weather and Climate Dynamics, 2021_

## Author Comment (AC2)

**Response to reviewer 2**

Review of "Dynamical drivers of Greenland blocking in climate models"

The present manuscript explores the deriver of Greenland blocking events using a set of multi-model large ensembles. This is a nice way to analyze blocking events as they are rare events. The authors showed that two (MIROC5 and ECHAM6.3-LR) out of five models fairly capture the frequency of blocking events compared to the ERA-Interim reanalysis. Then they used these two models to understand the main driver of Greenland blocking events and found an inconsistency in the dynamics of blocking events between the models. More specifically, the author found a link between CWB and Greenland blocking events in ECHAM6.3-LR model in contrast to MIROC5, which CWB is largely underestimated. They further expand their analysis to future climate and report the changes in RWB and eddy-driven jet to investigate the connection with blocking frequency.

I found the topic and contents relevant, and the paper is falling within the scope of Weather and Climate Dynamics. I have no major comments, and I strongly suggest that for publication after some minor revisions, which I believe are needed to improve the quality of the paper and make it clear.

The authors thank the reviewer for the detailed review and constructive comments. Our point-bypoint responses can be found in blue here below.

L7: The authors mentioned "four out five models" but then throughout the paper, they focused on the two models which they believe have better performance compared to the other models. Even in conclusion (L324) they mentioned "three out of five" and I suggest they change it in the abstract to be consistent with the analyses in the paper.

We agree and will modify the abstract in order to mention only the two models ECHAM6.3-LR and MIROC5.

L42: "have shown"

Will be corrected.

L85: I think it sounds better if the authors rephrase the sentence and mentioned 100 and 501 members instead of "times"

We agree and we will change the sentence accordingly.

L112: Shouldn't it be "Blocking grid points are identified" instead of "Blocking events are identified" as the index identifies each grid point that satisfies the condition, which does not necessarily mean they are separate events.

Yes, we will now use "Blocked grid points" instead of "Blocking events" as suggested by the reviewer, which is also now in better agreement with another sentence, further down, where we also used "grid points".

L115: I am unsure if I can follow the time fraction (%) definition that the authors explained here. Since it is an important point, they need to explain that clearly. Is the fraction defined as (total blocked gridpoints/(total gridpoints\*number of days))?

We agree that the wording was confusing and we will change the sentence.

L116: Have the authors thought about performing clustering analysis to separate Greenland blocks instead of subjective criteria to count the number of blocked days as I think the Greenland region will show up naturally as a region of frequent blocking events base on what the authors have shown in Fig. 1(f).

Despite recent improvements in weather regimes representation by high-resolution climate models (Fabiano et al. 2020), models with lower resolution struggle in representing the weather regimes (Dawson et al., 2012). An alternative would have been to project the geopotential anomalous field on the Greenland anticyclone regime obtained from ERA-Interim but, for consistency, the clustering should be performed with the 10 years or 9 winters of ERA-Interim, which is a short period to obtain reliable regimes. Moreover, this would have attenuated the differences in the blocking pattern itself among the models. In addition, it is not evident that the clustering analysis would give the same Greenland blocking events as the detection method used here. Therefore, we opted for a simple blocking index.

L133: "GB" spell it out

Yes, thank you for spotting it.

L166: Over Greenland, negative bias in the frequency of blocking is not significant in the ECHAM6.3-LR model in addition to MIROC5, and it needs to be added. Also, in the following sentence (L168) the author mentioned "strong negative biases in the ensemble mean." which I think is not true for those two models over Greenland based on what has been shown in Fig. 1.

The reviewer is correct and we will change the sentence accordingly.

L211: I believe that ECHAM6.3-LR is the only one that has the smallest mean-state bias in Fig. 3, and biases in MIROC5 are the same as the other models (specially CanAM4 seems to be even better compared to MIROC5).

To clarify this sentence, we have represented the standard deviation taken at 30°N (south of the mean jet position) for all models (see Fig. R1 hereafter). This figure shows that MIROC5 and ECHAM6-3-LR are the models that performs the best compared to ERA-Interim, especially between 40°W and 20°W. We will add this figure to the Supplement and refer to it in the main text.

L247: Fig. 5d-f not Fig. 5c-e

Will be corrected.

L269: Can weak absolute vorticity gradient be why the frequency of blocking events is captured with less bias in MIROC5? If yes, I think the authors need to expand this part.

We will add more details about the study of Luo et al. (2019) such as: "In an idealized set-up, Luo et al. (2019) showed that a weak mean meridional gradient of potential vorticity at high latitudes

Figure R1: Ensemble mean of the DJF mean daily standard deviation of the 850-hPa zonal wind at  $30^{\circ}$ N (units: m s-1) as a function of the longitude (covering the North Atlantic domain) for every HAPPI ensemble. The green line shows the DJF mean daily standard deviation for ERA-Interim (2006-2015).

leads to reduced energy dispersion, enhanced nonlinearity, and more persistent eddy forcing, favouring long and intense blocking. Even though MIROC5 does not exhibit more intense or longer blocking than ECHAM6.3-LR, this mechanism could also trigger blocking thus enhancing its frequency."

L275: Fig. 6m-o not Fig. 5l-n

Will be corrected.

L279: I think the authors mean Fig. S8 instead of Fig. S9.

Yes, it will be corrected.

L293: I understand the numbers are not significant compared to previous studies but are the changes significant compared to each models' current climate?

In this sentence, we actually mean that the changes in blocking frequency over Greenland between the present and future experiments are weak and not significant. In addition, the values are also lower than the values found in previous studies such as shown in the recent paper of Davini and D'Andrea (2020) because the warming between the HAPPI experiments is of  $1.2^{\circ}$ C, which is lower than the warming compared to the pre-industrial in the CMIP5/6 experiments (+2.6 to 4.8°). We will make this part clearer.

L297: It is unclear what the authors refer to by saying "large decrease". Isn't it a weak decrease as they found here, and it is mentioned in L293?

Yes, the decrease is not large. We will remove the word "large".

Last comment: Captions of figures are vague and ambiguous, making it hard for readers to follow. They should be clear as much as possible that readers understand the figures just by reading the captions. Also, if the anomalies are significant (by any t-test) please mentioned that in the caption, otherwise please do statistical significance analysis for those cases that anomalies are shown.

We will improve the captions of the figures and add the information about the significance (e.g. t-test or bootstrap). We will also perform a significance analysis for the composites (Figs. 4, 5, and 6) using a bootstrap method. The lack of significance to a 10% level will be shown with dots. From first calculations, it seems that the anomalies in the composites are all significant (see e.g. Fig. R2).

---

## Author Comment (AC3)

**Response to reviewer 3**

Review of manuscript wcd-2021-26:

Dynamical drivers of Greenland blocking in climate models

by

Clio Michel et al.

In this study, the HAPPI large ensemble simulations are used to investigate the representation of Greenland blocking (GB) in the HAPPI models and the extent to which GB occurrence is associated with the occurrence of Rossby wave breaking, specifically cyclonic Rossby wave breaking (CWB) over the Labrador Sea / the south of Greenland (70-30W, 50-70N). It is found that most of the 5 HAPPI models considerably underestimate GB frequency; the best agreement with ERA-Interim GB frequency is seen in ECHAM6 and especially the MIROC5 models. Models also tend to underestimate CWB frequency, especially so MIROC5. An analysis of the relationship between CWB and GB occurrence shows that these two phenomena are closely associated in ERA-Interim and ECHAM6, but much less so in MIROC5, leading the authors to conclude that a different dynamical driver of GB must be acting in MIROC5.

Global climate models (GCMs) have long-standing biases in the representation of atmospheric blocking, which are still seen in the latest generation of GCMs. Studies into the nature of blocking (biases) in GCMs therefore remain important, and the present study illustrates an intriguing difference between two GCMs in how they represent GB and CWB. I therefore recommend the study for publication in WCD. I would like to make one suggestion for further analysis, other than that my comments are minor/presentational.

We are grateful to the reviewer for the careful review. Our answers to the comments are below in blue color.

1)

The authors conclude that the dynamical link between CWB and GB is present but not the main ingredient triggering GB in MIROC5. This is, of course, very interesting and nicely illustrated in the paper but the question about the character of GB in MIROC5 is left open. Given the large negative CWB bias in MIROC, many blocking events will occur without CWC in that model.

I would like to suggest the following additional analysis: The authors have derived a set of days (or events) when CWC and GB occurs, respectively. This could be used to derive  $2x^2$  contingency tables of co-occurrence of CWC and GB with frequency counts (No-No, No-Yes, Yes-No, Yes-Yes) for each of ECHAM, MIROC, and ERA-Interim. Furthermore, composites could then be shown with respect to three sets of these events (No-Yes, Yes-No, Yes-Yes), and potentially these could be more illustrative than the current Figures 5 and 6 – it is difficult to be sure, but I would like to encourage the authors to try if they have not already done so.

We thank the reviewer for the suggestion of this additional analysis that we have performed and whose results are described below.

In order to define the Greenland blocking category, we use the Greenland blocking days as defined in the manuscript. The no CWB category contains the days when the CWB index defined in the manuscript is equal to 0 and the CWB category includes all days with a positive CWB index. The contingency Table R1 clearly shows that, for MIROC5, Greenland blocking (GB) occurs as frequently with CWB as without CWB (51 versus 50 days), whereas for ECHAM6.3-LR and ERA-Interim most of Greenland blocking occurs with CWB. This difference may arise from the lack of CWB in MIROC5. However, the composites of 500-hPa geopotential for the category GB - no CWB (first row column (b) in Figs. R1, R2, and R3) exhibits a westward shift of the anticyclonic pattern. This may reflect the blocking at a later stage of its lifetime as recently shown by Drouard et al. (2021) for the blocking of cyclonic type typical over Greenland (with CWB southwest of the region chosen for the index calculation as seen on Fig. R3c second row but only for ERA-Interim). Whether or not CWB occurs during Greenland blocking, the low-level zonal wind is always stronger south over the North Atlantic (columns (a) and (b) of fourth row in Figs. R1, R2, and R3).

We think that the composites in the original manuscript and this additional analysis complement each other: Fig. 4 of the original manuscript shows the atmospheric state during Greenland blocking days, Fig. 5 of the original manuscript demonstrates that Greenland blocking can happen with CWB southwest of Greenland, and the above analysis nicely illustrates that MIROC5 has many days with Greenland blocking without CWB in the box chosen. Therefore, we have decided to include the contingency table in the manuscript but to leave out the composites that will be added to the supplement.

Table R1: Ensemble mean and spread of the number of days in each category for ECHAM6.3-LR and MIROC5 (2nd and 3rd columns) and total number of days in each category for ERA-Interim (4th column). Unit: days. The CWB/no CWB categories distinguish the days for which the spatially averaged CWB frequency, as defined in Section 2.4 of the original manuscript, is greater than 0 / equals 0. GB stands for Greenland Blocking. The GB/no GB categories distinguish the blocked days from the non-blocked days as defined in Section 2.3 of the original manuscript.

|       | ECHAM6.3-LR     |                | MIROC5          |                  | ERA-Interim |        |
|-------|-----------------|----------------|-----------------|------------------|-------------|--------|
|       | CWB             | no CWB         | CWB             | no CWB           | CWB         | no CWB |
| GB    | $36.2 \pm 14.9$ | $10.0\pm6.0$   | $51.2 \pm 13.6$ | $49.8 \pm 15.2$  | 77          | 26     |
| no GB | $431.9\pm20.5$  | $333.9\pm22.6$ | $323.4\pm20.1$  | $385.6 \pm 24.4$ | 406         | 303    |

---

## Author Response (AR1)

**Response to reviewer 1**

Review of "Dynamical drivers of Greenland blocking in climate models" by Clio Michel et al.

This manuscript investigates the drivers of atmospheric blocking in the Greenland region using several large ensembles of climate models. The representation of blocking and other dynamical features in each of the models is compared to ERA-Interim. Based on this analysis two models (ECHAM6.3-LR and MIROC5) are selected based on their performance in simulating historical blocking in the region of interest. The study continues to investigate several atmospheric features such as Rossby wave breaking and and wind speed in present and future in order to identify the drivers behind the expected changes in winter blocking in the Greenland region. Interestingly, the authors find that different dynamical features seem to be driving blocking in the two models considered.

This is relevant and timely work on the underlying dynamics driving atmospheric blocking for one of the main blocking regions in the northern hemisphere. I think the study fits well into the scope of Weather and Climate Dynamics and generally I think it should be published and I have no major comments. However, I found the manuscript to contain a large number of smaller issues, inconsistencies, and ambiguities that made it very hard to follow the authors points and sometimes even impossible to be sure what the authors were referring to. I would urge the authors to carefully address these and particularly focus on a consistent description of their methodology as well as their figures.

*We thank the reviewer for the careful review of the manuscript and detailed comments. Our point-by-point responses are in blue here below. We will carefully revise the manuscript with a particular focus on the description of the methodology.*

Comments

35: CMIP actually stands for Coupled Model Intercomparison Project as far as I know

*True! Corrected.*

84: As additional information the author could state how much warming that is approximately compared to the present decade used in the paper.

*The sentence has been changed as follows: "In this study, we are interested in the present decade which covers 2006-2015 and a future decade in which the global annual mean temperature is $2°C$ warmer than the pre-industrial level ($\sim +1.2°C$ compared to the present decade).".*

Figure S1: I have to admit I've trouble interpreting that figure for several reasons:
- The caption is unclear to me: "Difference of blocking frequency where blocking is detected". Do both panels show blocking frequency or some kind of difference?
- I assume that the figure actually shows absolute blocking frequencies and not any kind of difference. So to show if there are "substantial changes in the results" (line 114 of the main manuscript) the author should plot differences (or even relative difference even though I realize that his might be problematic as well).

- It is unclear to me what the data-basis for this plot is. 'All-Hist' in the title lets me assume that it refers to all 125 members of NorESM1 in the historical period but this does not seem to be the case as the blocking frequency looks too patchy for that – so is it possible that it is only one member?
- The high 'blocking' frequency in the Atlantic below 40 degree north is not normally defined as blocking (at least not in the classical sense), right? The authors focus on blocking over Greenland so I would suggest optimizing the colorbar to the blocking frequency there.

Figure R1 below (Fig. S1 of the supplement) represents the DJF mean blocking frequency for the first member of the present-day experiment only. Following the reviewer's suggestion, we have added a third panel to the figure that shows the difference between the two blocking frequencies (see Fig. R1 below). Moreover, the colorbar of the top two panels has been adjusted to better highlight the blocking frequency north of 40°N, the unit is added to the figure, and the caption has been rewritten to make it clearer and add the third panel's description.

[Figure]

Figure R1: DJF mean blocking frequency (unit: % of winter days) for member 1 of the NorESM1-HAPPI ensemble, with blocking detected using the 500-hPa geopotential on (a) the model grid and (b) a grid of 2° × 2°. (c) Difference between the two mean frequencies (a) minus (b) where (a) has been remapped on the same 2° × 2° grid as (b).

Section 2.3: Did the authors calculate the annual blocking frequency and then select Greenland blocking days in DJF from that or was the blocking frequency already calculated only for DJF? In the second case the first and last 4 days of each season will have a lower probability of blocking due to the 5 day persistence criterion I assume?. This should be made clear.

*We detected blocking with a 5-day persistence using the whole 10 years of each member. Then, we selected the days occurring during the 9 winters (January, February 2006 and December 2015 are excluded to only keep full winter seasons) such that there is no issue with the first and last 4 days of each DJF season. We have rewritten this part of the method section as follows:* "*Winter-time blocking climatologies are obtained by averaging all blocked grid points over time (excluding January, February 2006 and December 2015 to only keep full DJF seasons) and are expressed as a percentage of the number of winter days ($90 \times 9 = 810$ days).*".

133: I assume GB stands for Greenland blocking?

*Yes, changed to* "*Greenland blocking*".

134: Why is there a difference between the blocking and the anticyclone definition? Should blocking not just be a stationary anticyclone?

*Yes, blocking can be considered as a stationary anticyclone. However, the blocking detection method used in this study relies on the overturning of a geopotential contour, which does not necessarily happen when there is an anticyclone. That is why we use an objective detection algorithm that is identical to a cyclone detection except that it detects highs in the 500-hPa geopotential anomaly field instead of lows in the mean sea level pressure field. Looking at the anticyclone frequency can tell us if there are anticyclones over Greenland that are not linked to an overturning of a geopotential contour and without any minimum persistence. We have added the following short paragraph in the method section (2.5) to better justify our use of this anticyclone detection:* "*We use this objective detection algorithm because even though a blocking can be considered as a stationary anticyclone, an anticyclone can occur without reversal of geopotential contours, which is the method used in this study to detect blocking. Thus, we are able to see if there are anticyclones over Greenland that are not linked to an overturning of a geopotential contour and without any minimum persistence.*"

150: Figure S2 does not show that "none of the 30 climatologies of 9 consecutive winters [...] is significantly different from the total 40-year [...] climatology". as the authors seem to indicate here it just shows the blocking climatologies. How was this established? Was that done on a grid-point level or somehow averaged? Would we not even expect about 10% of cases to be outside of the 90% range?

*As written in the caption of Fig. S2, we conducted a t-test that assessed the difference between each of the 9-winter blocking frequency climatologies and the 40-year (or actually 39-winter) climatology of the blocking frequency. Therefore, each grid point is tested and nowhere the two climatologies were found to be significantly different.*

Figure 1:
- I personally think that there should be an indication of what is shown as well as the units directly in the figure and not just in the caption.
- "Black lines show the 2, 4 and 6% contours for ERA-Interim (2006-2015)." For me these lines appear grey but that might just be my viewer.
- In panel f the black/grey contours do not seem to match the respective shading. This might just be due to different interpolation but it maybe should be clarified.
- I assume that the white areas without dots just have zero blocking frequency (rather than significant

biases)?

The units have been added next to the colorbar not only to Fig. 1 but also on all the other figures of the revised manuscript and supplement. The contour lines are indeed grey. The contours do not match the shading because they have been smoothed. Yes, the white areas without dots do not have blocking. All these details have been added to the caption.

165: Could the authors rephrase this? I did not count but it looks to me as if for all models there are more dotted than un-dotted gird cells even when only focusing on 'blue' areas so I don't think the concluding that "Most of the negative biases are significant" (except for MIROC5) is supported as it is.

In addition to the reviewer's comment, there was a mistake in Fig. 1 so we have changed the sentence to: *"All models exhibit significant (non-dotted) negative biases over Greenland and UK, with MIROC5 having also a significant positive bias southwest of Greenland. MIROC5 is the model with the smallest bias and ECHAM6.3-LR the model with the second lowest bias over Greenland."*

168: This seems to partly contradict the earlier statement that 9 years are enough to establish a robust blocking climatology, right? (even though the argument there was only made for observations). I would argue that for a sufficiently long climatological period the blocking frequency should be a model property and not depend on the realization, right?

As written by the reviewer, we showed for ERA-Interim that the overlapping 9-winter blocking frequency climatologies are not significantly different from the 39-winter climatology. This was performed to justify the fact that we will use ERA-Interim's period 2006-2015 (the period of the simulations) to calculate the biases. In line 168, we look at the ensemble spread. Therefore, it is not surprising that some members better reproduce ERA-Interim's climatology than others. Earlier at the end of Section 2.1, we state that the use of large ensembles overcomes the fact that nine winters might be too short to obtain realistic blocking frequency climatologies, even though Davini and D'Andrea (2016) found that nine winters should be enough to get relatively accurate blocking statistics.

Figure 2: I am again not sure what is shown in this figure. The caption suggest to me that it shows the area average of blocking frequency over the GB region: "mean Greenland blocking frequency" but this is not correct, right? Frequencies are too high for that, so I assume it actually shows Greenland blocking days as defined in line 116? Please clarify.

The figure shows the percentage of Greenland blocking days during the nine winters of each decade, hence a Greenland blocking frequency. The caption has been rewritten as follows: *"Distributions of the nine-winter (DJF) mean Greenland blocking frequency (in 2% bins) for each ensemble. The mean frequency of each model is shown in the title and given in Table 1. Shown in every panel is the mean Greenland blocking frequency from ERA-Interim for 2006-2015, which is 12.68% (black line), and the minimum/maximum blocking frequencies from nine-consecutive winters for the whole ERA-Interim period of 1979-2018, which are 5.66 and 14.16% respectively (dashed lines). Gray bars show the number of members with no GB blocking in the nine-year period."*.

Figure S3: The unit of the shading should be somewhere in this figure.

We have included the unit of the shading in both the figure and the caption.

Figure 3: I assume that for ERA-Interim not the ensemble mean but the mean over the different periods is shown?

True, the caption has been rewritten as follows: *"Ensemble means of the DJF mean daily standard deviation of the 850-hPa zonal wind (shading, in m s$^{-1}$) and of the DJF 850-hPa zonal wind climatology (contours, interval: 3 m s$^{-1}$, zero contour omitted, negative values with dashed lines) for (a) CAM4-2degree, (b) CanAM4, (c) ECHAM6.3-LR, (d) MIROC5, and (e) NorESM1-HAPPI. The daily standard deviation is calculated for each member separately and then averaged over the ensemble. (f) DJF mean daily standard deviation (shading, in m s$^{-1}$) and climatology (contours, interval: 3 m s$^{-1}$, zero contour omitted, negative values with dashed lines) of the 850-hPa zonal wind for the period 2006-2015 of ERA-Interim.".*

211: "MIROC5 and ECHAM6.3-LR are the models with the largest variability on the equatorward side of the mean jet and the smallest mean-state biases with respect to wind variability." Can the authors go into a bit more detail here? Looking at figure 3 I am not sure if I can see what the authors mean with this statement. In particular the standard deviation from MIROC5 does not seem to be higher southwards of its climatological jet than any other models'.

[Figure]

Figure R2: Ensemble mean of the DJF mean daily standard deviation of the 850-hPa zonal wind at 30°N (units: m s$^{-1}$) as a function of the longitude (covering the North Atlantic domain) for every HAPPI ensemble. The green line shows the DJF mean daily standard deviation for ERA-Interim (2006-2015).

We meant that the larger standard deviation of the zonal wind at 850 hPa shows that the eddy-driven jet can shift south of its climatological position more often for ECHAM6.3-LR than MIROC5. Figure R2 represents more clearly that the models with the largest standard deviation of the 850-hPa zonal wind south of the mean jet at 30°N and most similar to ERA-Interim are ECHAM6.3-LR and MIROC5. This figure has been added in the supplement. The sentence has been changed to: *"MIROC5 and ECHAM6.3-LR are the models with the largest variability on the equatorward side of the mean jet (30°N, Fig. S6) hence the smallest bias in wind variability.".*

216: "Climatologies show that AWB is more frequent and located on the equatorward side of the jet at both low and upper levels (solid contours in Fig. 4 left)" What is meant by "low and upper levels"

and where is this shown in figure 4? This sentence somehow seems to suggest that the solid contours in figure 4 are an indication of the jet but to know the position relative to the jet the reader actually needs to compare figure 4 to figure 3f, right?

*The sentence is indeed not clear and has been changed to:* *"The ERA-Interim climatology of RWB frequency shows that AWB is most frequent on the equatorward side of the mean jet (compare red contours to gray shading in Fig. S7f) while CWB is less frequent than AWB but shows a maximum frequency on the poleward side of the mean jet (compare blue contours to gray shading in Fig. S7f) (see also Martius et al. 2007)."*.

Section 3.3: It would be interesting to discuss these results in the light of this recent study as well: https://onlinelibrary.wiley.com/doi/10.1029/2020JD034082

*See response to comment about Figure S8a below.*

l251: "ECHAM6.3-LR [...] slightly underestimates the Greenland blocking frequency" I'm not sure if I would call a mean underestimation of about 50% (if I read figure 2c correctly) 'slightly'.

*The reviewer is right if we consider Fig. 2c (or Table 1). However, Fig. 1c shows an underestimation of the ensemble mean Greenland blocking frequency of -2 to -3%. As Fig. 1c is the type of figure usually shown in the literature dealing with blocking frequency and where a 'more regular' blocking detection is used (not using the blocking index defined in this study), we refer to Fig. 1c in this sentence. Therefore, the sentence has been changed to:* *"Of the five models examined here, ECHAM6.3-LR is the least biased in terms of mean state, variability, and RWB, and the Greenland blocking frequency is only underestimated by 2-3% as seen on Fig. 1c".*

Figure S8a: Again it would be interesting to see if the results are consistent with Drouard et al. (2021) when split by AWB and CWB.

*We thank the reviewer for this comment. Several studies, including the recent Drouard et al. (2021), showed or mentioned that Greenland blocking are driven mainly by CWB. Therefore, we do not think that splitting the blocking events in CWB and AWB type would yield additional insight into the problems addressed in the present manuscript. Also, previous studies showed that Greenland blocking lasts longer than other blocking (e.g. Scandinavian blocking), in accordance with the study of Drouard et al. (2021), a fact that we now mention in the introduction.*

291: CMIP5 was published around 2013 if I'm not mistaken. Can the authors check if the cited studies really all look at CMIP5 data?

*Indeed, the studies of Sillmann and Croci-Maspoli (2009) and Barnes et al. (2012) used CMIP3 simulations and we have added the CMIP6 results as well:* *"In agreement with previous studies using CMIP3, CMIP5, and CMIP6 experiments (e.g., Sillmann and Croci-Maspoli, 2009; Barnes et al., 2012; Masato et al., 2013; Woollings et al., 2018; Davini and D'Andrea, 2020) [...]".*

292: "we note a decrease in the ensemble mean blocking frequency over Greenland, in particular for ECHAM6.3-LR (up to -0.6% of the time)..." Is this referring to Greenland blocking days as defined in line 116 or to blocking frequency directly as I would assume from this statement?

*The percentages refer to the responses (Future minus Present experiments) of the ensemble mean blocking frequency as seen over Greenland in Fig. R3, figure that is now included in the supplement and cited in the main text. The figure shows the ensemble mean percentage of blocking days relative*

to the total number of winter days (90*9=810 days). We have removed the confusing "of the time" in the parenthesis but added the reference to the figure. The sentence now reads: *"[..], we note a weak and non-significant decrease between the present and future experiments in the percentage of blocked days (see Table 1) and in the ensemble mean blocking frequency over Greenland, in particular for ECHAM6.3-LR (up to -0.5%, Fig. S15c) and MIROC5 (up to -1.5%, Fig. S15d)."*

[Figure]

Figure R3: Ensemble mean response (shading, difference between the future and present experiments) of the DJF mean blocking frequency (percentage of blocking days relative to the total number of winter days, in %) for the five HAPPI models. The contours show the ensemble mean DJF mean blocking frequency (first contour and interval: 1%).

295: It seems to me that not the change relative to pre-industrial is important but the change relative to the historical period?

We refer here to the pre-industrial period as the this is the reference for the +2°C target. Following this comment, we noticed that the temperature increases given for the end of the century were relative to the historical period 1986-2005 and not the pre-industrial 1850-1900. We have changed the sentence to keep the reference of the pre-industrial period: *"This decrease is weaker compared to the studies cited above (e.g., -2 to -4% over Greenland in the CMIP multi-model mean responses in Fig. 6a-c of Davini and D'Andrea, 2020) mainly because the HAPPI future experiments represent a very mitigated warming scenario with a global mean temperature increase of +2°C relative to pre-industrial climate compared to the +3.2 to 5.4°C at the end of the 21st century for the Representative Concentration Pathway 8.5 of CMIP5 (IPCC, 2013)."*

302: "The composites over the blocked days are very similar between the present and future experiments (Fig. 7)" Figure 7 does not show the present?

*The sentence is indeed not clear. The reader has to compare Fig.7 to the left and middle columns of Fig. 5. We have added this information.*

Figure S11: I assume this figure shows change between historical and future?

*Yes, the caption has been changed accordingly: "Ensemble mean responses (difference between the future and present experiments) of Rossby wave breaking frequency (blue contours for CWB and red contours for AWB, solid lines for positive values and dashed lines for negative values, zero-contour omitted, interval: 0.5%) and 850-hPa zonal wind (black contours and gray shading, unit: m s$^{-1}$) for DJF for the five HAPPI models."*

311: "Over the North Atlantic, CWB do not seem to change much (very weak values and noisy field; not shown)" What does 'not shown' mean in this context? CWB is shown in figure 8 and S11, right? There just happen to be no contours in the frame for the case of figure 8b? Or do I misunderstand something? What does 'seem to change' mean? In a given model it either changes or not, right?

*As guessed by the reviewer, the response of the CWB frequency is small and does not appear with the contour interval chosen. We have changed the text to: "Over the North Atlantic, the CWB frequency does not change (noisy field with amplitudes below first contour level) [...]".*

336: Except for AWB in MIROC5, right?

*Correct, the sentence is changed to: "Rossby wave breaking patterns are quite well represented by most models, MIROC5 being the exception, but there is still a negative bias for both AWB and CWB almost everywhere in the European-North Atlantic domain and a positive bias of AWB over the Mediterranean.".*

Technical comments:
42: 'have showed'
Figure 6: "(g,h,i) Same as (c,d) but for anticyclonic wave breaking frequency"
288: "CWB are"

*Thank you, they are all corrected.*

**Response to reviewer 2**

Review of "Dynamical drivers of Greenland blocking in climate models"

The present manuscript explores the deriver of Greenland blocking events using a set of multi-model large ensembles. This is a nice way to analyze blocking events as they are rare events. The authors showed that two (MIROC5 and ECHAM6.3-LR) out of five models fairly capture the frequency of blocking events compared to the ERA-Interim reanalysis. Then they used these two models to understand the main driver of Greenland blocking events and found an inconsistency in the dynamics of blocking events between the models. More specifically, the author found a link between CWB and Greenland blocking events in ECHAM6.3-LR model in contrast to MIROC5, which CWB is largely underestimated. They further expand their analysis to future climate and report the changes in RWB and eddy-driven jet to investigate the connection with blocking frequency.

I found the topic and contents relevant, and the paper is falling within the scope of Weather and Climate Dynamics. I have no major comments, and I strongly suggest that for publication after some minor revisions, which I believe are needed to improve the quality of the paper and make it clear.

*The authors thank the reviewer for the detailed review and constructive comments. Our point-by-point responses can be found in blue here below.*

L7: The authors mentioned "four out five models" but then throughout the paper, they focused on the two models which they believe have better performance compared to the other models. Even in conclusion (L324) they mentioned "three out of five" and I suggest they change it in the abstract to be consistent with the analyses in the paper.

*We agree and have modified the abstract in order to mention only the two models ECHAM6.3-LR and MIROC5:* *"We focus on two models that both underestimate CWB frequency but with different representation of the wintertime Greenland blocking frequency. Nevertheless, they both show the typical Greenland blocking features [...]"*

L42: "have shown"

*Thank you. Corrected.*

L85: I think it sounds better if the authors rephrase the sentence and mentioned 100 and 501 members instead of "times"

*We agree and we have changed the sentence accordingly:* *"This multi-model ensemble comprises five atmospheric general circulation models (AGCM) with between 100 and 501 members for each period.".*

L112: Shouldn't it be "Blocking grid points are identified" instead of "Blocking events are identified" as the index identifies each grid point that satisfies the condition, which does not necessarily mean

they are separate events.

*Yes, we now use "Blocked grid points" instead of "Blocking events" as suggested by the reviewer, which is also now in better agreement with another sentence, further down, where we also used "grid points".*

L115: I am unsure if I can follow the time fraction (%) definition that the authors explained here. Since it is an important point, they need to explain that clearly. Is the fraction defined as (total blocked gridpoints/(total gridpoints*number of days))?

*We agree that the wording was confusing and we have changed the sentence to: "Winter-time blocking climatologies are obtained by averaging all blocked grid points over time (excluding January, February 2006 and December 2015 to only keep full DJF seasons) and are expressed as a percentage of the number of winter days ($90\times9=810$ days).".*

L116: Have the authors thought about performing clustering analysis to separate Greenland blocks instead of subjective criteria to count the number of blocked days as I think the Greenland region will show up naturally as a region of frequent blocking events base on what the authors have shown in Fig. 1(f).

*Despite recent improvements in weather regimes representation by high-resolution climate models (Fabiano et al., 2020), models with lower resolution struggle in representing the weather regimes (Dawson et al., 2012). An alternative would have been to project the geopotential anomalous field on the Greenland anticyclone regime obtained from ERA-Interim but, for consistency, the clustering should be performed with the 10 years or 9 winters of ERA-Interim, which is a short period to obtain reliable regimes. Moreover, this would have attenuated the differences in the blocking pattern itself among the models. In addition, it is not evident that the clustering analysis would give the same Greenland blocking events as the detection method used here. Therefore, we opted for a simple blocking index.*

L133: "GB" spell it out

*Yes, thank you for spotting it.*

L166: Over Greenland, negative bias in the frequency of blocking is not significant in the ECHAM6.3-LR model in addition to MIROC5, and it needs to be added. Also, in the following sentence (L168) the author mentioned "strong negative biases in the ensemble mean." which I think is not true for those two models over Greenland based on what has been shown in Fig. 1.

*As there was a mistake in Fig. 1, the biases are now significant for all models so we have changed the sentences to "All models exhibit significant (non-dotted) negative biases over Greenland and UK, with MIROC5 having also a significant positive bias southwest of Greenland. MIROC5 is the model with the smallest bias and ECHAM6.3-LR the model with the second lowest bias over Greenland." and "Accurate Greenland blocking can occasionally be reproduced by a few members of some models even though these models exhibit negative biases in the ensemble mean.".*

L211: I believe that ECHAM6.3-LR is the only one that has the smallest mean-state bias in Fig. 3, and biases in MIROC5 are the same as the other models (specially CanAM4 seems to be even better compared to MIROC5).

*To clarify this sentence, we have represented the standard deviation taken at 30°N (south of the mean jet position) for all models (see Fig. R1 hereafter). This figure shows that MIROC5 and*

ECHAM6-3-LR are the models that performs the best compared to ERA-Interim, especially between 40°W and 20°W. We have added this figure to the supplement and refer to it in the main text: *"MIROC5 and ECHAM6.3-LR are the models with the largest variability on the equatorward side of the mean jet (30°N, Fig. S6) hence the smallest bias in wind variability."*.

[Figure]

Figure R1: Ensemble mean of the DJF mean daily standard deviation of the 850-hPa zonal wind at 30°N (units: m s$^{-1}$) as a function of the longitude (covering the North Atlantic domain) for every HAPPI ensemble. The green line shows the DJF mean daily standard deviation for ERA-Interim (2006-2015).

L247: Fig. 5d-f not Fig. 5c-e

Thank you. Corrected.

L269: Can weak absolute vorticity gradient be why the frequency of blocking events is captured with less bias in MIROC5? If yes, I think the authors need to expand this part.

We have added more details about the study of Luo et al. (2019): *"Luo et al. (2019) showed in an idealised set-up that, at high latitudes, a weak mean meridional gradient of potential vorticity, associated with weak mean wind, leads to reduced energy dispersion, enhanced nonlinearity, and more persistent eddy forcing, favouring long and intense blocking. Even though MIROC5 does not exhibit more intense or longer blocking than ECHAM6.3-LR, this mechanism could also trigger blocking thus enhancing its frequency."*

L275: Fig. 6m-o not Fig. 5l-n

Thank you. Corrected.

L279: I think the authors mean Fig. S8 instead of Fig. S9.

*Yes, it is corrected but it is now Fig. S10.*

L293: I understand the numbers are not significant compared to previous studies but are the changes significant compared to each models' current climate?

*In this sentence, we actually mean that the changes in blocking frequency over Greenland between the present and future experiments are weak and not significant. In addition, the values are also lower than the values found in previous studies such as shown in the recent paper of Davini and D'Andrea (2020) because the warming between the HAPPI experiments is of 1.2°C, which is lower than the warming compared to the pre-industrial in the CMIP5/6 experiments (+3.2 to 5.4°). We have tried to make this part clearer: "we note a weak and non-significant decrease between the present and future experiments in the percentage of blocked days (see Table 1) and in the ensemble mean blocking frequency over Greenland, in particular for ECHAM6.3-LR (up to -0.5%, Fig. S15c) and MIROC5 (up to -1.5%, Fig. S15d). This decrease is weaker compared to the studies cited above (e.g., -2 to -4% over Greenland in the CMIP multi-model mean responses in Fig. 6a-c of Davini and D'Andrea, 2020) mainly because the HAPPI future experiments represent a very mitigated warming scenario with a global mean temperature increase of +2°C relative to pre-industrial climate compared to the +3.2 to 5.4°C at the end of the 21st century for the Representative Concentration Pathway 8.5 of CMIP5 (IPCC , 2013)."*

L297: It is unclear what the authors refer to by saying "large decrease". Isn't it a weak decrease as they found here, and it is mentioned in L293?

*Yes, the decrease is not large. We have remove the word "large": "Previous studies showed that the decrease in Greenland blocking frequency [...]".*

Last comment: Captions of figures are vague and ambiguous, making it hard for readers to follow. They should be clear as much as possible that readers understand the figures just by reading the captions. Also, if the anomalies are significant (by any t-test) please mentioned that in the caption, otherwise please do statistical significance analysis for those cases that anomalies are shown.

*We have tried to improve the captions of the figures and added the information about the significance (e.g. t-test or bootstrap). We have performed a significance analysis for the composites (Figs. 5, 6, and 7) using a bootstrap method. The lack of significance to a 10% level is shown with dots on all figures. We have added the method details in the methods section.*

**Response to reviewer 3**

Review of manuscript wcd-2021-26:

Dynamical drivers of Greenland blocking in climate models

by

Clio Michel et al.

In this study, the HAPPI large ensemble simulations are used to investigate the representation of Greenland blocking (GB) in the HAPPI models and the extent to which GB occurrence is associated with the occurrence of Rossby wave breaking, specifically cyclonic Rossby wave breaking (CWB) over the Labrador Sea / the south of Greenland (70-30W, 50-70N). It is found that most of the 5 HAPPI models considerably underestimate GB frequency; the best agreement with ERA-Interim GB frequency is seen in ECHAM6 and especially the MIROC5 models. Models also tend to underestimate CWB frequency, especially so MIROC5. An analysis of the relationship between CWB and GB occurrence shows that these two phenomena are closely associated in ERA-Interim and ECHAM6, but much less so in MIROC5, leading the authors to conclude that a different dynamical driver of GB must be acting in MIROC5.

Global climate models (GCMs) have long-standing biases in the representation of atmospheric blocking, which are still seen in the latest generation of GCMs. Studies into the nature of blocking (biases) in GCMs therefore remain important, and the present study illustrates an intriguing difference between two GCMs in how they represent GB and CWB. I therefore recommend the study for publication in WCD. I would like to make one suggestion for further analysis, other than that my comments are minor/presentational.

We are grateful to the reviewer for the careful review. Our answers to the comments are below in blue color.

1)

The authors conclude that the dynamical link between CWB and GB is present but not the main ingredient triggering GB in MIROC5. This is, of course, very interesting and nicely illustrated in the paper but the question about the character of GB in MIROC5 is left open. Given the large negative CWB bias in MIROC, many blocking events will occur without CWC in that model.

I would like to suggest the following additional analysis: The authors have derived a set of days (or events) when CWC and GB occurs, respectively. This could be used to derive 2x2 contingency tables of co-occurrence of CWC and GB with frequency counts (No-No, No-Yes, Yes-No, Yes-Yes) for each of ECHAM, MIROC, and ERA-Interim. Furthermore, composites could then be shown with respect to three sets of these events (No-Yes, Yes-No, Yes-Yes), and potentially these could be more illustrative than the current Figures 5 and 6 – it is difficult to be sure, but I would like to encourage the authors to try if they have not already done so.

We thank the reviewer for the suggestion of this additional analysis that we have performed. We think that the composites in the original manuscript and this additional analysis complement each other: Fig. 4 of the original manuscript shows the atmospheric state during Greenland blocking days, Fig. 5 of the original manuscript demonstrates that Greenland blocking can happen with CWB southwest of Greenland, and the above analysis nicely illustrates that MIROC5 has many days with Greenland blocking without CWB in the box chosen. Therefore, we have decided to include the contingency table R1 (here below) in the manuscript but to leave out the composites (Figs. R1, R2, and R3) that have been added to the supplement.

We have added a new paragraph to the Discussion section: *"Table 2 shows that Greenland blocking occurs as frequently with CWB (GB-CWB) as without CWB (GB-no CWB) for MIROC5 (51.2 versus 49.8 days), whereas for ECHAM6.3-LR and ERA-Interim Greenland blocking occurs most frequently with CWB (36.2 versus 10.0 days for ECHAM6.3-LR and 77 versus 26 days for ERA-Interim). This difference probably arises from the lack of CWB in MIROC5. Also, the composites of the 500-hPa geopotential for the category GB-no CWB exhibits a westward shift of the anticyclonic anomaly compared to the GB-CWB category (see first row in Figs. S11, S12 and S13). This may reflect the blocking at a later stage of its lifetime as recently shown by Drouard et al. (2021) for the blocking of cyclonic type typical over Greenland. Whether or not CWB occurs during Greenland blocking, the low-level zonal wind is always stronger south over the North Atlantic (see columns (a) and (b) of the fourth row in Figs. S11, S12 and S13)."*.

Table R1: Ensemble mean and spread (standard deviation over the members) of the number of days in each category for ECHAM6.3-LR and MIROC5 and total number of days in each category for ERA-Interim. Unit: days. The CWB/no CWB categories distinguish the days for which the spatially averaged CWB frequency, as defined in Section 2.4, is greater than 0 / equals 0. GB stands for Greenland Blocking. The GB/no GB categories distinguish the blocked days from the non-blocked days, as defined in Section 2.3. The sum of the number of days in the four categories for each model and ERA-Interim equals the total number of winter (DJF) days in the decade 2006-2015.

|  | ECHAM6.3-LR | | MIROC5 | | ERA-Interim | |
|  | CWB | no CWB | CWB | no CWB | CWB | no CWB |
|---|---|---|---|---|---|---|
| GB | $36.2 \pm 14.9$ | $10.0 \pm 6.0$ | $51.2 \pm 13.6$ | $49.8 \pm 15.2$ | 77 | 26 |
| no GB | $431.9 \pm 20.5$ | $333.9 \pm 22.6$ | $323.4 \pm 20.1$ | $385.6 \pm 24.4$ | 406 | 303 |

[Figure]

Figure R1: Ensemble mean of the composites of the 500-hPa geopotential (Z500), AWB, CWB, 850-hPa zonal wind (U850) and blocking for three categories (GB - CWB, GB - no CWB, and no GB - CWB) whose mean numbers of days are found in the contingency table for ECHAM6.3-LR. The first four top rows shows the anomalies in shading and the total field in contours and the bottom row shows the blocking frequency in shading. The numbers in the bottom right corner of each panel show the number of members that have days in the category. Here all members have at least one day in all categories. GB stands for Greenland Blocking.

[Figure]

Figure R2: As Fig. R1 but for MIROC5.

[Figure]

Figure R3: Composites of the 500-hPa geopotential (Z500), AWB, CWB, 850-hPa zonal wind (U850) and blocking for three categories (GB - CWB, GB - no CWB, and no GB - CWB) whose numbers of days are found in the contingency table for ERA-Interim. The first four top rows shows the anomalies in shading and the total field in contours and the bottom row shows the blocking frequency in shading. The number 1 in the bottom right corner of each panel points out that this is ERA-Interim and not an ensemble. GB stands for Greenland Blocking.

2) abstract
Make clear early on that this study is about the winter season.

We have added this information at two places in the abstract.

3) page 1, line 23
Surely there are also more wide-ranging impacts of GB due to its association with the NAO and temperature anomalies across much of the Northern Hemisphere? Please add a brief discussion and some references; Chen et al. 2017 looks like a good start.

*We thank the reviewer for the reference suggestion. We have modified the first paragraph of the introduction to include this aspect: "Moreover, Greenland blocking has been shown to cause melting events of the Greenland Ice Sheet (Fettweis et al., 2013; McLeod and Mote, 2015; Hermann et al., 2020; Hanna et al., 2021), by mainly reducing the cloud cover, increasing temperatures (Chen and Luo, 2017), and changing the surface energy budget (Ward et al., 2020, and references therein), which can impact global sea level rise (Van den Broeke et al., 2016). In addition to the local impact, Greenland blocking is also associated with temperature anomalies over the whole Northern Hemisphere (Chen and Luo, 2017)."*.

4) page 3, line 63
What does "enhanced" mean here? With respect to which reference? Maybe it can simply be omitted.

*As suggested, we have removed the word "enhanced" there.*

5) page 4, line 121
Typo: occur*s*

*Thank you. Corrected.*

6) page 5, line 152
If I understand correctly, these 30 estimates are from overlapping time periods and therefore strongly dependent. Is this considered in the significance testing?

*No, the dependency between the short period and the whole 40 years was not taken into account with this t-test. In order to avoid this shortcoming, we now calculate the standard deviation around the 2006-2015 mean for ERA-Interim by bootstrapping 100 times 30 non-consecutive winters (with replacement) within 1979-2018. Because of a mistake in our original figure, we now see that the blocking frequency biases are significant in most regions with non-zero values, even for MIROC5 and ECHAM6.3-LR over Greenland. We now use Fig. R4 in the revised manuscript and Fig. 4 representing the RWB biases has been changed as well. The text has been changed accordingly.*

7) Section 3
It is nice to see from this section and the Supplement that the authors have conducted thorough model evaluation. A question that keeps coming up is what constitutes a "good enough" evaluation result to justify pursuing the main aim of the study. Where the authors clear about this before conducting the evaluation? I understand this question is hard and slightly philosophical, but, if possible, a short discussion of this point with respect to this study would be greatly appreciated!

*Our purpose is not to understand the processes leading to blocking in the real world, for which we would have had to define a criterion in advance, evaluate the models, and throw out the models failing to meet the criterion. Our goal is rather to evaluate the biases in the models, understand the reasons for these biases hence providing a more in-depth model evaluation compared to what has been published previously. Therefore, we first evaluate the biases and investigate the processes in each.*
*Many studies have evaluated blocking bias in climate models, but only few tried to understand the reasons behind the biases. Here, we highlight that such reasons might depend on the considered models. We hope that the results will help the modellers to design numerical experiments and consider*

[Figure]

Figure R4: (a-e) Bias in winter (DJF) blocking frequency for the five models (ensemble mean of the blocking frequency minus ERA-Interim) and (f) ERA-Interim DJF blocking climatology for 2006-2015 (in frequency, as %). Black Dark gray lines show the smoothed 2, 4 and 6% contours for ERA-Interim (2006-2015). The black box shows the main region of Greenland blocking in ERA-Interim. Biases that are not significant at the 10% significance level are dotted and there is no dot where there is no blocking.

these dynamical aspects and not only the mean state when they create/evaluate/tune the models.

We have slightly changed the last paragraph of the conclusion to make clearer that we focus on explaining the biases and not the processes leading to blocking: *"Our study highlights that, in order to evaluate blocking representation in climate models, we should not just consider biases in the mean state. It is also important to evaluate the representation of the known mechanisms that lead to blocking, such as CWB, which is an indicator for the eddy-mean flow interaction. Davini et al. (2017) started to tackle this issue by studying the representation of eddies in one climate model with various spatial resolutions, finding that higher resolution simulations do not necessarily better represent eddies. A better understanding of the biases sources in the mechanisms leading to blocking in climate models is crucial to reduce those biases and improve the prediction of future changes."*

8) page 7, line 208

"Similar standard deviation ..." – similar to ERA or to each other?

*We meant similar to ERA-Interim. The sentence has been changed to: "All HAPPI models exhibit standard deviation values similar to ERA-Interim, however, only on the poleward side of the climatological jet between the southern tip of Greenland and Iceland.".*

9) page 7, line 215

"More frequent ..." than what? Rephrase this sentence to make clearer that you are talking about a model bias.

*We agree with the reviewer and have made the sentence clearer: "The ERA-Interim climatology of RWB frequency shows that AWB is most frequent on the equatorward side of the mean jet (compare red contours to gray shading in Fig. S7f) while CWB is less frequent than AWB but shows a maximum frequency on the poleward side of the mean jet (compare blue contours to gray shading in Fig. S7f) (see also Martius et al., 2007)."*

10) Table 1

Given the central role of CWB, I suggest adding columns for CWB frequency to this table.

*As far as we understand this suggestion, we have calculated the ensemble mean of the CWB index for each model (all winter days and all members are taken into account). The index represents the area of the box covered by CWB and a value of 100% would mean that every grid point in the box has CWB. All values can be found in Table R2 here below. As expected from the bias shown in Fig. 4 of the original manuscript, MIROC5 exhibits low values for the present and future experiments but ECHAM6.3-LR exhibits values close to ERA-Interim's. We have added the table values to Table 1 of the manuscript and cite them in the text at two places: in Section 3.3 and in the introduction of Section 4.*

*Table R2: Ensemble mean of the DJF CWB index (in %) as defined in Section 2.4 for the five HAPPI models and both present and future experiments along with the DJF mean of the CWB index for ERA-Interim (2006-2015).*

| Model/Reanalysis | Experiment | Mean CWB index |
|---|---|---|
| CAM4-2degree | Present | 7.6 |
| CanAM4 | Present | 10.1 |
| ECHAM6.3-LR | Present | 11.6 |
| MIROC5 | Present | 4.7 |
| NorESM1-HAPPI | Present | 8.4 |
| CAM4-2degree | Future | 7.4 |
| CanAM4 | Future | 10.4 |
| ECHAM6.3-LR | Future | 11.5 |
| MIROC5 | Future | 4.7 |
| NorESM1-HAPPI | Future | 8.2 |
| ERA-Interim | 2006-2015 | 11.1 |

11) page 9, line 237

"zonal wind" – I would add the direction ("westerly").

Added.

12) Section 4.2
Please consider if some of the supplemental material referred to could be promoted to the main manuscript.

We would like to give the readers the opportunity to see the results from all the models, but would like to keep the focus in the paper on the comparison between ECHAM6.3-LR and MIROC5. Therefore, most of the results for the other models are included only in the supplement. However, we will have now some additional information in the main manuscript in form of tables: the contingency table R1 and the numbers of Table R2.

13) page 10, line 291
Please add the CMIP6 results, which are now also available (Davini & D'Andrea, 2020).

Added: *"In agreement with previous studies using CMIP3, CMIP5, and CMIP6 experiments (e.g., Sillmann and Croci-Maspoli, 2009; Barnes et al., 2012; Masato et al., 2013; Woollings et al., 2018; Davini and D'Andrea, 2020) [...]"*

14) page 11, line 305
Please add appropriate figure cross-reference(s) here (Figures 5 & 7, I think).

Added: *"(compare Fig. 7 to the left and middle columns of Fig. 5)"*

15) page 11, line 330
"for the reasons cited above" ... please refer to the (sub)section here and/or add a brief reminder of what these reasons are.

The reasons are stated in the two sentences before this one. We have anyway added the reasons in parenthesis in this sentence as suggested by the reviewer.

16) Section 6
Please add a brief summary/conclusions from Section 5.

Yes, we have added a short paragraph to sum up the results of Section 5 in the conclusion: *"In agreement with previous studies, ECHAM6.3-LR and MIROC5 both exhibit a decreased frequency of Greenland blocking in the future experiments. However, this decrease is, here, not significant and not clearly linked to a reduced frequency of CWB, as could have been expected from previous studies (e.g., Barnes and Hartmann, 2012). Moreover, Greenland blocking composites of the geopotential, zonal wind, and RWB for the future period are very similar to the composites for the present period.".*

References

Chen, X., & Luo, D. (2017). Arctic sea ice decline and continental cold anomalies: Upstream and downstream effects of Greenland blocking. Geophysical Research Letters, 44(7), 3411–3419. https://doi.org/10.1002/2016GL072387

Davini, P., & D'Andrea, F. (2020). From CMIP3 to CMIP6: Northern Hemisphere Atmospheric Blocking Simulation in Present and Future Climate. Journal of Climate, 33(23), 10021–10038. https://doi.org/10.1175/JCLI-D-19-0862.1

---

## Author Response (AR2)

**Response to reviewer 1**

2nd review of "Dynamical drivers of Greenland blocking in climate models" by Clio Michel et al.

Thank you to the authors for carefully addressing my comments from the first round of revisions. I have only one new issue on the significance testing for composites which I think is slightly flawed as presented. Otherwise and as written before I think the manuscript should be published in Weather and Climate Dynamics as it fits well into the scope of the journal and presents very timely work.

> Thank you for your second careful review of the manuscript.

Minor comments:

section 2.6.2: Given that blocked/unblocked days are highly auto-correlated, the bootstrapping should account for this by not drawing X random days buy picking random periods corresponding to the blocked/unblocked periods in the original time series. See e.g., methods of Brunner, L., Hegerl, G. C., & Steiner, A. K. (2017). Connecting atmospheric blocking to European temperature extremes in spring. Journal of Climate, 30(2), 585–594. https://doi.org/10.1175/JCLI-D-16-0518.1

> We thank the reviewer for the thoughtful suggestion. We have calculated again the significance following Brunner et al. (2017) as seen with the dots in Figs. 5, 6, and 7. The new calculation highlighted a small mistake in the calculation of the events duration, which lead us to change Fig. S10 of the supplement as well. However, the results do not change much and the conclusions are the same. The method as been updated in Section 2.6.2 (see lines 170-173).

section 2.3/Figure 2: I think the usage of blocking frequency is still ambiguous. The convention to define the fraction of blocked days in the Greenland region as 'Greenland blocking frequency' seems somewhat counter-intuitive. Without careful reading of the methods the readers interpretation of figure 2 might easily be that it shows the average blocking frequency over Greenland (so, e.g., the average of blocking in the black box in figure 1f for ERA-I – which is NOT what is shown). My suggestion would be to clearly distinguish between the two, e.g., by using 'frequency of blocked days' instead of 'blocking frequency' as xlabels in figure 2.

> We agree and have changed the x-axis labels on Fig. 2. We have also changed the captions of Fig. 2 and Table 1 accordingly.

**Response to reviewer 3**

Review of manuscript wcd-2021-26 (version 2):

Dynamical drivers of Greenland blocking in climate models

by

Clio Michel et al.

I reviewed the original submission and have now read the second version of the manuscript as well as the author responses. I would like to thank the authors for additional analyses and discussion, some of which in response to my earlier comments about compositing, significance testing, and the evaluation of model biases. Although some questions must remain open, one very interesting aspect of this study is to have a specific example of two models that show very similar Greenland blocking (GB) frequency, but a very different representation of dynamical fields thought to be closely associated with GB – cyclonic Rossby wave breaking in this case. This example will surely be an interesting reference for future research. I recommend this study for publication in WCD with (nearly) no further comments:

The authors thank the reviewer for his second review of our manuscript. Our response to the only comment is in blue here below.

Figure 6:
The caption says "... (g,h,i) Same as (c,d) but for anticyclonic wave breaking frequency. ..." Is this correct?

It is not correct. It has been changed to *"(g,h,i) Same as (d,e,f) but for anticyclonic wave breaking frequency."*. Thank you for spotting the mistake.